# Objects guide human gaze behavior in dynamic real-world scenes

**Nicolas Roth**[1,2]*, **Martin Rolfs**[1,3,4], **Olaf Hellwich**[1,5], **Klaus Obermayer**[1,2,4]

**1** Cluster of Excellence Science of Intelligence, Technische Universität Berlin, Germany, **2** Institute of Software Engineering and Theoretical Computer Science, Technische Universität Berlin, Germany, **3** Department of Psychology, Humboldt-Universität zu Berlin, Germany, **4** Bernstein Center for Computational Neuroscience Berlin, Germany, **5** Institute of Computer Engineering and Microelectronics, Technische Universität Berlin, Germany

* roth@tu-berlin.de

**Data Availability Statement:** The code can be found in this github repository: https://github.com/rederoth/ScanDy. All data and the results of the evolutionary parameter optimization can be found

## Abstract

The complexity of natural scenes makes it challenging to experimentally study the mechanisms behind human gaze behavior when viewing dynamic environments. Historically, eye movements were believed to be driven primarily by space-based attention towards locations with salient features. Increasing evidence suggests, however, that visual attention does not select locations with high saliency but operates on attentional units given by the objects in the scene. We present a new computational framework to investigate the importance of objects for attentional guidance. This framework is designed to simulate realistic scanpaths for dynamic real-world scenes, including saccade timing and smooth pursuit behavior. Individual model components are based on psychophysically uncovered mechanisms of visual attention and saccadic decision-making. All mechanisms are implemented in a modular fashion with a small number of well-interpretable parameters. To systematically analyze the importance of objects in guiding gaze behavior, we implemented five different models within this framework: two purely spatial models, where one is based on low-level saliency and one on high-level saliency, two object-based models, with one incorporating low-level saliency for each object and the other one not using any saliency information, and a mixed model with object-based attention and selection but space-based inhibition of return. We optimized each model's parameters to reproduce the saccade amplitude and fixation duration distributions of human scanpaths using evolutionary algorithms. We compared model performance with respect to spatial and temporal fixation behavior, including the proportion of fixations exploring the background, as well as detecting, inspecting, and returning to objects. A model with object-based attention and inhibition, which uses saliency information to prioritize between objects for saccadic selection, leads to scanpath statistics with the highest similarity to the human data. This demonstrates that scanpath models benefit from object-based attention and selection, suggesting that object-level attentional units play an important role in guiding attentional processing.

in this OSF repository: https://www.doi.org/10.17605/OSF.IO/83XUC.

**Funding:** This work was funded by the Deutsche Forschungsgemeinschaft (DFG, German Research Foundation) under Germany's Excellence Strategy – EXC 2002/1 "Science of Intelligence" – project number 390523135 to M.R, O.H and K.O. M.R. was supported by the Heisenberg program of the Deutsche Forschungsgemeinschaft (DFG grants RO3579/8-1 and RO3579/12-1). The funders had no role in study design, data collection and analysis, decision to publish, or preparation of the manuscript.

**Competing interests:** The authors have declared that no competing interests exist.

## Author summary

There has long been an interest in understanding how we decide when and where to move our eyes, and psychophysical experiments have uncovered many underlying mechanisms. Under controlled laboratory conditions, objects in the scene play an important role in guiding our attention. Due to the visual complexity of the world around us, however, it is hard to assess experimentally how objects influence eye movements when observing dynamic real-world scenes. Computational models have proved to be a powerful tool for investigating visual attention, but existing models are either only applicable to images or restricted to predicting where humans look on average. Here, we present a computational framework for simulating where and when humans decide to move their eyes when observing dynamic real-world scenes. Using our framework, we can assess the influence of objects on the model predictions. We find that including object-based attention in the modeling increases the resemblance of simulated eye movements to human gaze behavior, showing that objects play indeed an important role in guiding our gaze when exploring the world around us. We hope that the availability of this framework encourages more research on attention in dynamic real-world scenes.

## Introduction

Humans actively move their eyes about 2–3 times per second when visually exploring their environment [1, 2]. This moves the part of the retina with the highest visual acuity—and allocates significant computational capacities in the brain—to parts of the scene that are of the highest interest at a given time. Hence, the decision of where and when to move our eyes determines which parts of the world around us we inspect in more detail and, conversely, which parts we might miss. Moving our eyes is therefore always a trade-off between different potential target positions that fundamentally shapes the way we perceive the world. The sequence of eye movements for a given visual scene is called a scanpath and reflects the allocation of overt attention (in contrast to covert attention, i.e., the ability to attend to parts in the visual field without moving the eyes). Most knowledge about overt (and covert) visual attention stems from highly controlled psychophysical experiments, which are carefully designed to isolate and examine a single attentional mechanism. Thus, these experiments only provide limited information on the extent to which these mechanisms actually guide our decisions on where to move our eyes in complex dynamic scenes like real-world videos. In this work, we present a modular, mechanistic computational framework to bridge the gap between attentional mechanisms unveiled under standardized conditions and their effect on *where* and *when* we move our eyes when observing ecologically valid scenes. We use this framework to quantitatively investigate the influence of object-based attention, which has been extensively investigated experimentally (e.g., [3–5]; for reviews see [6, 7]), in contrast to space-based attention on human gaze behavior in dynamic real-world scenes.

Computational models of visual attention in free-viewing conditions have proven to be a powerful tool for understanding human exploration behavior [8, 9]. Existing models of human eye movements, however, typically suffer from at least one of two major simplifications: the restriction to static scenes, and the static nature of the predicted behavior as an average spatial distribution of fixation locations rather than modeling eye movement patterns of individual observers. Scanpath prediction in static scenes was pioneered by the seminal work of Ref. [10], who implemented the previously postulated concept of a saliency map [11] algorithmically and suggested a strategy to sequentially select locations in the saliency map based on a

"winner-take-all" and a subsequent "inhibition of return" mechanism. A more detailed model of attentional dynamics and saccadic selection in static scene viewing was proposed with the *SceneWalk* model family [12–15]. These models predict the likelihood of moving the gaze to a certain position based on a foveated access to saliency information and a leaky memory process. Using a similar mathematical approach, Ref. [16] model the decision for choosing an upcoming saccade target as a selection process governed by either local or global attention. The currently best-performing scanpath prediction model for static scenes, DeepGaze III [17], uses a deep neural network for extracting high-level image features and combines them with the fixation history to predict the most likely fixation target.

The predicted outputs of these scanpath models for static scenes are ordered lists of $(x, y)$ fixation locations for a given image. The saccade timing, i.e., when to switch from one predicted fixation location to the next, is typically not of interest. Similarly, the standard evaluation metrics for scanpaths only consider predicted fixation locations but do not take the duration of fixations into account [18]. In dynamic scenes, the image content at a given $(x, y)$ location also depends on the time, and the scanpath must be specified as a list of $(x, y, t)$ locations and requiring prediction of saccade timing. Although the temporal aspect of scanpaths is much less investigated, there is preliminary evidence that variability in fixation duration depends on both cognitive factors and visual features within the scene [19, 20]. In the framework proposed here, a saccade is performed as soon as a potential target in the saccadic decision-making process has accumulated sufficient evidence. Similarly, Ref. [21] model the decision-making mechanisms that account for saccade timing and programming as a random-walk process in the CRISP model which, however, does not make any prediction about the $(x, y)$ location of the saccade target.

A few models exist that simultaneously account for when and where the eyes move when exploring static scenes. Ref. [22] argue for the need for models that combine the spatial and temporal aspects of gaze behavior and present the WALD-EM model. They introduced a spatiotemporal likelihood function to statistically model fixation positions and fixation durations, which are assumed to follow a WALD (inverse Gaussian) distribution, simultaneously and were successful in reproducing many aspects of scanpath statistics. The most recent version of the SceneWalk model [15] also incorporates fixation durations when generating scanpaths. The distribution of fixation durations is assumed to follow a Gamma distribution, which is parameterized such that the mean fixation duration at a location parametrically depends on the local image saliency at this position. As in the WALD-EM model [22], fixation durations are then included in the likelihood function, allowing for Bayesian inference of the model parameters. This approach could, in theory, be extended to scanpaths simulation in dynamic scenes, but updating the saliency map for every frame leads to a number of practical implications that make it computationally infeasible. In the LATEST model [23], each pixel-location of an image accumulates evidence in favor of moving the gaze from the current position to this location depending on scene features (like salience and semantic interest) and oculomotor factors (like saccade amplitude or change in direction). Being a statistical model, each respective influence on saccade timing is estimated from human scanpaths using generalized linear mixed models, without focusing on explaining the biological mechanisms that influence these factors. The location that is quickest to reach sufficient evidence consequently becomes the saccade target, resulting in a simultaneous prediction of where and when to move the gaze position. Predicting the saccade timing and target location simultaneously is especially important in dynamic scenes, where the scene content at a given pixel location can change over time. The evidence accumulation process in LATEST, a linear approach to threshold with ergodic rate [24], however, does not generalize to situations when scene features change during the accumulation process.

Apart from the fact that the designs of existing models (in their current form) are not able to process dynamic scenes, scene changes also lead to qualitatively different gaze behaviors. The presence of motion leads to a decrease in between-subject variability [25–27], and motion is the most important independent predictor for where people tend to look [28–30]. Dynamic scenes also give rise to a type of eye movement that models have yet to account for, smooth pursuit, where the eyes remain fixated on a moving object. Modeling scanpaths in dynamic scenes hence requires a new class of models that account for the location and timing of saccadic decisions, smooth pursuit eye movements, and scene changes as a result of object or camera movement.

Existing models for visual attention in *dynamic scenes* do not attempt to model the actual visual exploration behavior. Instead, they predict frame-wise saliency maps, quantifying the average spatial distribution of human gaze locations. Dynamic saliency models can be implemented by extending static saliency models to include motion features such as flicker and oriented motion energy. These motion conspicuity maps can be combined with maps of other low-level features to create a single saliency map for each frame [29, 31, 32]. Ref. [33] suggest that the best way of integrating motion into a static, low-level visual saliency model is by spatiotemporal filtering of the image intensity, inspired by the temporal receptive field of simple cells in V1. With the availability of larger datasets in recent years [34, 35], video saliency detection has also become a popular task in computer vision. Deep neural network (DNN) architectures, which include the temporal information in videos either through temporal recurrence [36, 37] or by using 3D convolutional networks [38–40], clearly outperform mechanistic models from computational neuroscience and psychology in predicting where humans tend to look. This boost in performance can be explained by the capabilities of these networks not just to encode information on low-level features like color or edges. Instead, the DNNs, which are trained on human eye tracking data, typically use pre-trained features from object detection or action recognition tasks and learn to extract high-level features characterizing objects and scene information during the learning process. Despite the remarkable performance of deep learning approaches in predicting where humans tend to look on average, the insights such models provide about the visual system are limited. Interpreting DNNs is notoriously hard and crucial information about how individuals decide to explore the scene is lost when averaging the viewing behavior of multiple observers.

A related task to video saliency detection is egocentric gaze prediction [41–43], which aims at predicting gaze position recorded with a mobile eye tracker from the first-person perspective. The field of view of the video data changes depending on the body and head movement of the participant, which is typically not predicted in this task. Despite being ecologically relevant, egocentric gaze prediction is therefore limited in the sense that for each video stimulus, there is only one ground truth scanpath, and that gaze is only one component of the exploration behavior, with head and body movement already provided by the field of view of the camera.

The role of visual objects in guiding human attention has been widely discussed in the literature. In the absence of task instructions, it has long been thought that the targets of eye movements are selected based on space-based attention, approximated by the bottom-up saliency of simple visual features [8, 10]. In the literature on mechanistic scanpath models it is still the dominant view that saccade targets are selected through space-based attention on the basis of saliency maps [14, 16]. However, there is growing evidence that objects also play an important role for the selection of saccade targets in scene viewing [44–46]. It has been established in a wide range of psychophysical experiments that objects influence visual attention in multiple ways (e.g., [4, 5, 47], see [6, 7] for reviews, and Sec. *Objects as perceptual units* in *Methods: ScanDy* for more details). This inspired several computational models of visual attention that incorporate or predict object-based information. Ref. [48] describe how units of visual

information, so-called proto-objects, can be formed based on a saliency map. Other models segment proto-objects based on coherent image regions [49] or perform perceptual grouping [50, 51] and use these representations to improve saliency estimations in images. Ref. [52] present a machine vision system based on object-based attention, which predicts attentional shifts on synthetic and natural images. Consistent with the experimental evidence for the importance of object-based attention, these object-based attention models typically report improvements compared to respective space-based modeling approaches. Due to fundamentally different model architectures and complexities, the comparability between object-based models and space-based baselines is, however, usually rather limited.

In this work, we propose a new modeling framework called *ScanDy* that is capable of simulating realistic **scan**paths for **dy**namic real-world scenes, including saccade timing and smooth pursuit behavior. To achieve this, we utilize the output of video saliency models, combine it with the attentional dynamics of static scanpath models, adapt a saccadic decision-making model to make it applicable to dynamic scenes, and show that predicted scanpaths benefit from object-based attention and selection. *ScanDy* is a mechanistic modeling framework with a modular design. Human scanpaths are simulated based on the features of the scene, with visual sensitivity depending on the current gaze position and the previous scanpath history. In contrast to a deep learning approach, attentional mechanisms are implemented with a small number of well-interpretable parameters based on insights from psychophysical experiments. This allows the implementation of scanpath models, which differ in their assumptions about one or more mechanisms underlying gaze behavior, within the same framework, enabling a systematic and quantitative exploration of attentional mechanisms.

We demonstrate how hypotheses for scanpath simulation in dynamic real-world scenes can be tested using this framework. We focused on the role of objects for attentional guidance and how object and saliency information are combined. For this, we analyzed different model implementations within the *ScanDy* framework: The first model is purely space-based and uses a biologically plausible low-level saliency map (*S.ll* model). We compared this with an object-based model which uses the same low-level saliency map but where attentional mechanisms, like the computation of visual sensitivity, inhibition of return, and the saccade target selection (cf. Sec. *ScanDy* in Methods), are based on objects (*O.ll* model). Comparing only these two models may, however, be unfair due to the additional scene information (in the form of object segmentation masks) available to the object-based model. Therefore, we introduced a space-based model with a high-level saliency map (*S.hl* model) as an additional comparison, where object information may be contained in the scene features, but the attentional mechanisms acting on them are purely spatial. We also investigated an object-based model which does not use any saliency information but only the center bias (*O.cb* model) to better understand the interaction of object-based attention and saliency. Lastly, we explored the hypothesis of a spatial inhibition of return mechanism applying to locations in allocentric coordinates [53] together with an object-based visual sensitivity and target selection in a "mixed" model that uses the low-level saliency map (*M.ll* model). Model parameters were fitted such that the simulated scanpaths reproduce the saccade amplitude and fixation duration distributions of free-viewing eye tracking data on videos of the VidCom [54] dataset (cf. Sec. *Dataset* in Methods).

There are currently no established solutions for how to evaluate the model predictions for simulated scanpaths on dynamic scenes (cf. Sec. *Evaluation of simulated scanpaths* in Discussion). We assessed the model performance in out-of-training predictions concerning spatial and temporal fixation behavior, including the proportion of fixations detecting, inspecting, and returning to objects, as well as the proportion of fixations exploring the background (cf. Sec. *Functional scanpath evaluation* in Results). The comparison of different models within the

*ScanDy* framework shows that a model using object-based attention, which selects objects for foveal processing based on visual saliency (i.e., the *O.ll* model), produces exploration behavior that most closely resembles that of humans. We thus conclude that object-level attentional units play an important role in guiding attentional processing. The *ScanDy* framework, together with the code and data for reproducing our results, is open source and can be accessed on GitHub: https://github.com/rederoth/ScanDy.

## Methods

### *ScanDy*: A modular framework for scanpath simulation in dynamic real-world scenes

We introduce a new computational framework for **scanp**ath simulation in **dy**namic real-world scenes: *ScanDy*. It predicts the timing and the spatial positions of foveation events (fixation and smooth pursuit) and saccades. Saccade locations and timing are modeled through a sequential decision process between potential targets. The target selection depends on the spread of visual sensitivity around the current gaze position (Sec. *Visual sensitivity (II)*), the scene content quantified by the salience of the potential target (Sec. *Scene features (I)*), and the recent scanpath history (Sec. *Inhibition of return (III)*). The decision process of where and when the eyes should move is modeled with a drift-diffusion model (DDM, Sec. *Decision-making (IV)*) with one accumulation process for each potential saccade target. As soon as the decision threshold of a potential target is reached, a saccade is executed to this target (Sec. *Gaze update (V)*) with the saccade duration depending on the saccade amplitude [55], and the decision process is reset. In between saccades are foveation events, where the gaze position stays at the previously selected target. Depending on the motion of the foveated target in the visual field, these foveation events result in either fixation or smooth pursuit. During all foveation events, we model fixational eye movements [56–58] as an additive Brownian motion on top of the current gaze position. In the following, we describe the individual components of the framework and their respective realizations in purely space-based (Fig 1) and object-based (Fig 2) model variants. We then compare the predicted scanpaths obtained from the two model families, each with two different saliency representations, to evaluate our hypothesis that objects play an essential role in attentional guidance.

**Objects as perceptual units.** The space-based models are only influenced by objects in the scene as far as the saliency map captures object-related features. In the object-based models, on the other hand, objects are the fundamental building blocks for the implementation of attention mechanisms and modulate the *Visual sensitivity (II)*, *Inhibition of return (III)*, *Decision-making (IV)*, and *Gaze update (V)* processes. What constitutes a visual object is an open question [64, 65] and can strongly depend upon a given task or the scene content [66, 67]. In the context of object-based visual selection, there has to exist a pre-attentive neural representation of objects to allow for attention to be directed to them. These pre-attentive units for visual attention have been called proto-objects [48, 68]. The literature on unconscious object representations (cf. [69–71]) and unconscious object-based attention (cf. [72, 73]), however, suggests that the pre-attentive targets of object-based attention can be fully developed object representations with boundaries that do not need to be refined when attended [7]. Early work by Ref. [74] shows that humans are able to extract the layout of a novel scene and understand its meaning within less than 300 ms of a single fixation. Ref. [75] shows that the global structure of a scene, defined by meaningful object boundaries, can be inferred within the first 100 ms and suggests that perception is quickly organized around the inferred scene layout. In addition, eye tracking data across different tasks and types of scenes have shown that observers tend to direct their gaze towards the center of objects as defined by semantic categories rather

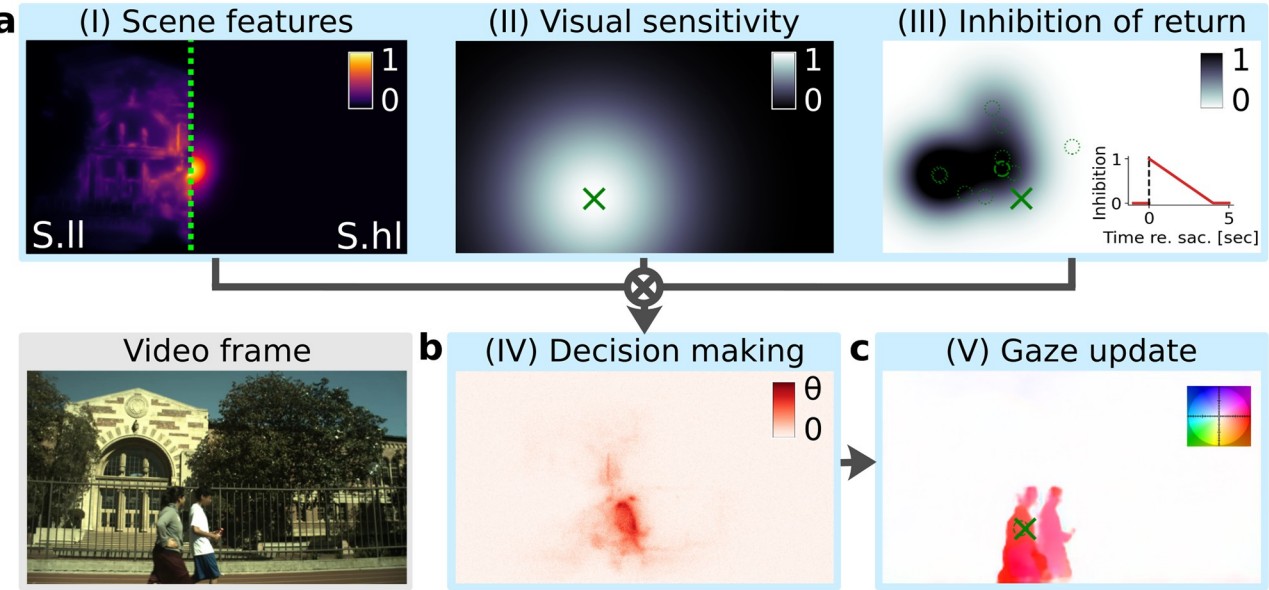

**Fig 1. Space-based scanpath model design.** Schematics of a space-based model within the modular *ScanDy* framework, illustrated for an example frame (*field03* of the VidCom dataset, see bottom left). (a) Modules (I-III) are computed simultaneously for each time step. The scene features (I) are quantified by a saliency map. For space-based models, we compared the impact of low-level saliency [33] (*left*) in model *S.ll* with high-level saliency as predicted by a deep neural network [38] (*right*, maps separated by the green dotted line) in model *S.hl*. The frame-wise saliency maps are multiplied with a generic center bias [59]. Color represents relative saliency, normalized by the highest saliency in the image. The visual sensitivity (II) is implemented as a Gaussian of width $\sigma_S$ centered at the current gaze point (green cross). The space-based inhibition of return mechanism (III) inhibits the previous target locations (shown as dotted circles, latest target (h = 1 in Eq (1)) as a dashed circle). A Gaussian of width $\sigma_I$ is centered around each previous saccade target location. The amplitude of each Gaussian linearly decreases relative to the time of the respective saccadic decision with slope $r$ (see inset). The inhibition map is the sum of weighted Gaussians around previous target locations, clipped to a maximum of 1. The output maps of these modules (I-III) are multiplicatively combined for the decision-making process (see Eq (5)). (b) The evidence for each potential saccade target (i.e., each pixel location) is accumulated in the decision variable (cf. Eq (5)) in module (IV). Each potential target is represented in a drift-diffusion model (DDM) where the drift rate is computed by combining values from maps of modules (I, *left*, model *S.ll*), (II) & (III), and the noise level $s$. (c) Module (V) updates the gaze position based on the resulting decision variables. If one pixel location in (IV) reaches the DDM decision threshold $\theta$, a saccade to this position is executed. Otherwise, the gaze position is updated (from dashed circle to cross) based on the optical flow calculated using PWC-Net [60] (plotted with flow field color coding in the top right where color indicates direction and hue indicates velocity). The parameters of the space-based model are listed in Table 1.

than towards preliminary proto-objects [44, 45, 76] and such semantic objects have been shown to predict gaze behavior better than low-level saliency [46]. Taken together, the experimental evidence suggests assuming semantic object segmentation masks as representations for object-based attention and selection. Hence, we computed the object masks based on semantics using the Mask R-CNN deep neural network architecture [77] as implemented in the Detectron 2 framework [61].

We used the VidCom dataset [54], which consists mainly of outdoor videos of people, animals, and traffic. Given this scene content, we defined humans, animals, and vehicles as objects and assigned the remaining portion of the scene to a general background category. We denote the object segmentation masks by $O_i$, $i \in [0, N_O]$, where $N_O$ is the number of objects appearing in a given video and $O_0$ is the segmentation mask of the background. Segmented objects were tracked across frames using DeepSort [78] and manually controlled for obvious segmentation or tracking mistakes. As argued for in Ref. [7], we assumed that object masks are fully evolved before being attended and do not depend on any visual information which could have been acquired during the scanpath. Future versions of *ScanDy* may, for comparison, have the option to start using coarse object representations, which will then be refined as visual information is

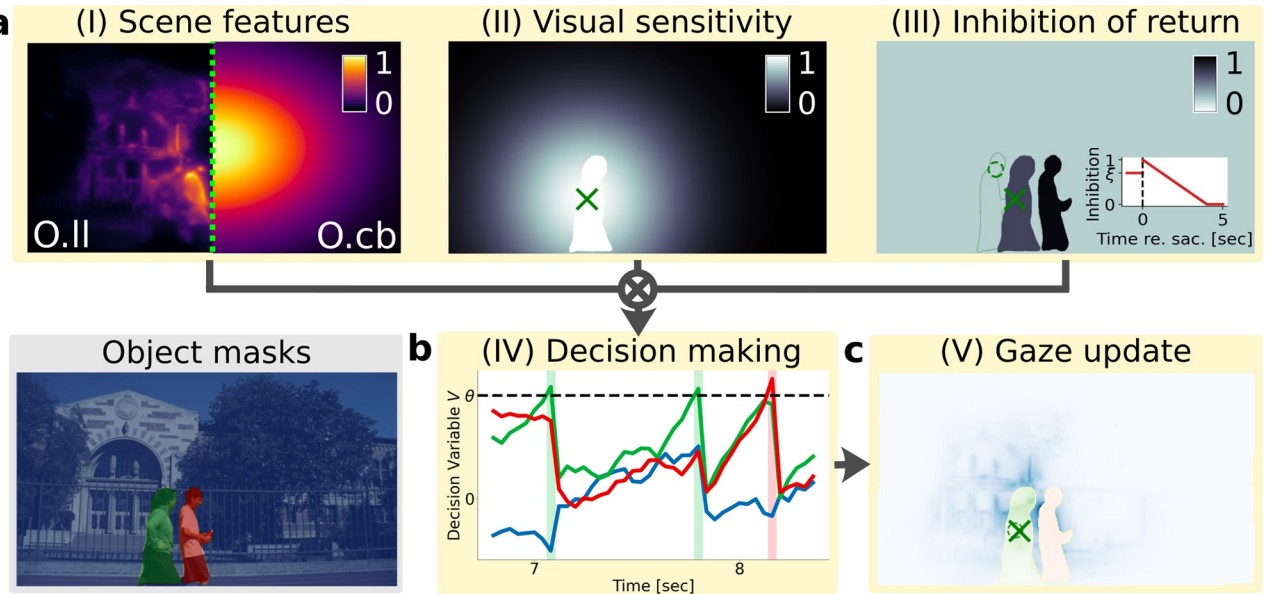

**Fig 2. Object-based scanpath model design.** Schematic of an object-based model within *ScanDy*, in analogy to Fig 1 and for the same video frame. The object masks (bottom left, 2 persons and background are shown) are based on a semantic segmentation using the Mask R-CNN deep neural network (implementation by [61]). (a) Any appropriate feature map could be used to encode the scene content in module (I). For the model comparison, we used the same low-level saliency [33] as in Fig 1 for the object-based model *O.ll* (I, *left*) but in a second implementation, model *O.cb*, did not include any kind of scene features and only used the generic center bias [59] (I, *right*). In addition to the space-based Gaussian of size $\sigma_S$, we account for a higher sensitivity across the currently foveated object [4, 62] by setting the sensitivity in module (II) across the foveated object mask to one. Instead of inhibiting locations, module (III) inhibits previously foveated object masks (cf. [63]). While foveated, an object is inhibited by value $\xi$ (cf. left object). As soon as the gaze position moves outside of the object mask, the inhibition is set to one (cf. right person), which then, over time, decreases again linearly to zero with slope $r$ (see inset). The scanpath history is identical to Fig 1 (III), with the dashed circle marking the previous saccade target location. At the time of the previous saccade, the right person was at the location of the green object contour, but in the meantime moved to the right and was followed with smooth pursuit before a saccade was initiated to the now foveated object (green cross). The output maps of modules (I-III) are again combined (see panel (c) for pixel-wise multiplication result of (I-III), with different color maps for each object), but the visual information is now summed across each object mask and normalized by the logarithm of the object size (see Eq (4)). The resulting value for each object is the drift rate for the drift-diffusion process (see Eq (5)). (b) In object-based target selection, the evidence for the saccadic decision-making is accumulated for each object in the scene, quantified by a DDM with threshold $\theta$ and noise level $s$ in module (IV). (c) The gaze position update (V) follows the movement of the foveated object mask (from dashed circle to cross). If the decision threshold is crossed, a saccade to the target object is executed. The exact landing position within the target object is probabilistic and proportional to the activity of the combined maps of modules (I-III) (see (7), combined maps shown in a different color for each object mask). The object-based models have the same number of free parameters as the space-based ones. They are listed in Table 2.

gathered during scene exploration. Other object classes—or object representations different from R-CNN computed masks—can also be included in the framework as long as the object masks are temporally consistent, meaning that an object appearing in consecutive frames has the same object ID $i$.

**Scene features (I).** The simulated eye movements are based on the visual information in the scene, which we quantify by frame-wise feature maps $F$. The default choice for $F$ is a bottom-up saliency computation based on motion, intensity, color, and orientation [33] (Fig 1a, I, *left*).

Alternatively, target relevance can be based on high-level features of the scene (Fig 1a, I, *right*). In this case, $F$ is computed with a deep neural network architecture called TASED-Net (Temporally-Aggregating Spatial Encoder-Decoder Network, [38]), which is trained on predicting the spatial distribution of fixation locations for each frame by using information from previous frames. As a result, TASED-Net prioritizes high-level scene elements, like faces, when compared to feature-based saliency models [33], and is highly sensitive to salient moving

objects [38], since it was implicitly trained to track and attend to objects. All saliency maps $F(x, y)$ used here were normalized to [0, 1] for better comparability.

When presenting complex scenes on a computer screen, observers tend to make fixations towards the center of the display. This is partially due to a tendency of interesting objects to be placed in the center during recording (*photographer bias*) [79, 80] but has been shown to persist irrespective of the scene content [59, 81]. Potential explanations for this scene-independent center bias are a learned expectation of interesting objects in the center (as a consequence of repeated exposure to photographer-biased stimuli), the maximal utilization of peripheral vision when looking at the center, a preference of humans for a straight-ahead position of the eyes, and fixational starting markers in the scene center in experimental paradigms combined with an oculomotor bias for short saccades [82]. Hence, we explicitly include this scene-independent center bias in our model. Following prior work [59, 83], we quantified the center bias as an anisotropic Gaussian $G_{cb}(x, y) = \exp\left(-\frac{(x-x_c)^2}{2\sigma_x^2} - \frac{(y-y_c)^2}{2\sigma_x^2 \cdot v}\right)$ with $\sigma_x^2 = 0.22$ (for x-values normalized to $-1 \leq x \leq 1$) and anisotropy ratio $v = 0.45$ and located at the image center $(x_c, y_c)$. All saliency maps in this work were multiplied with $G_{cb}$ to account for this central viewing bias.

**Visual sensitivity (II).**   In the foveated human visual system, visual acuity decreases with increasing eccentricity [84]. Hence, the extent to which certain features in a scene influence eye movements depends on the current gaze position $(x_0, y_0)$, for which we calculate a sensitivity map $S$ at each time step. We account for higher visual acuity in the fovea compared to the periphery with an isotropic Gaussian $G_S = \frac{1}{2\pi\sigma_S^2} \exp\left(-\frac{(x-x_0)^2+(y-y_0)^2}{2\sigma_S^2}\right)$. The standard deviation $\sigma_S$ is a free model parameter with a strong influence on the size of saccades in the simulated scanpaths.

Visual acuity is, however, just a first-order approximation of the actual sensitivity across our visual field. We can selectively attend to objects or locations [85, 86]. If humans are presented with a cue within an object, they respond faster to a target within the same object compared to a target outside of it, matched for the distance to the cue [4, 62]. This suggests a perceptual benefit of the currently attended object with an object-based distribution of spatial attention [6, 87]. In the object-based model, we approximate the spread of covert attention across a currently foveated object with a uniform sensitivity, replacing the part of the Gaussian $G_S$ that falls within this object.

This sensitivity map can be extended to also include covert and perisaccadic attentional shifts, as modeled by Ref. [14]. For simplicity and to keep the number of parameters low, the models described in the following do not include additional attentional mechanisms in the computation of the sensitivity map.

**Inhibition of return (III).**   Visual processing of stimuli, which have recently been the focus of attention, are often temporarily inhibited [88–90]. This inhibition of return (IOR) is commonly interpreted to encourage the exploration of previously unattended parts of the scene [91–93]. We adopt the common practice in scanpath models to use IOR in terms of an inhibition of previously explored parts of the scene for preventing the continuous return to the same targets with high saliency [8, 14, 94, 95].

The spatial extent of IOR has been shown to be highest at the attended location and graded with distance [96–99]. IOR in the space-based models is, therefore, implemented by a Gaussian-shaped inhibition $G_I(x, y)^{(h)} = \frac{1}{2\pi\sigma_I^2} \exp\left(-\frac{(x-x_h)^2+(y-y_h)^2}{2\sigma_I^2}\right)$ around the (pixel-)locations $(x_h, y_h)$ of previous saccade targets, where $h \in [1, N_F]$ denotes the index of previous saccade targets and $N_F$ the number of previous foveations. The time course of how IOR affects gaze behavior over time is, despite the vast literature on the effect, not well established [100]. For

simplicity, we assume that the effect decreases linearly with time. Hence, the space-based inhibition map $I$, bound between full ($I = 1$) and no inhibition ($I = 0$), at time $t$ is given by

$$I(x, y, t) = \min\left(1, \sum_{h=1}^{N_F} G_I(x, y)^{(h)} \cdot \max(0, (1 - (t - t_h)r))\right), \tag{1}$$

where $r$ determines the slope of the linear decrease with time and ($x_h, y_h, t_h$) the respective target location and time. If $r$ is close to zero, all $N_F$ previous foveation targets contribute to the inhibition, resulting in previously selected locations not being revisited. The standard deviation $\sigma_I$ of the Gaussian influences the average size of saccades and can, even though objects are not considered in this implementation, influence how often saccades are made within the same object.

Various paradigms suggest that IOR can be bound to objects in that it tracks and follows moving objects [63, 101–103]. In addition, IOR was found to spread across an object if attention was allocated to parts of the object [104, 105]. In the object-based models, we hence do not calculate the inhibition for each location as in Eq (1) but do so for each object in the scene. As long as an object is already foveated, meaning that the current gaze position ($x_0, y_0$) lies within the object segmentation mask $O_i$, the selection of this object for the next saccade is inhibited with the within-object factor $\xi \in [0, 1]$. To account for the typically much larger size and the heterogeneity of the background, we do not inhibit saccades within the background (within-object inhibition $\xi = 0$ for the background object). The inhibition for each object is set to one as soon as it is not foveated anymore and then linearly decreases with slope $r$ towards zero, as in the space-based IOR in (1). Consequently, the inhibition $I_i$ for an object at a given time is calculated as

$$I_i(t) = \begin{cases} \xi, & \text{if } (x_0, y_0) \in O_i \\ \max\left(0, 1 - r \cdot (t - t_h^{(i)})\right), & \text{else,} \end{cases} \tag{2}$$

where $t_h^{(i)}$ is the point in time when the gaze position was last on the object mask $O_i$.

There is, however, also experimental evidence for a reduction of IOR with object movement by Ref. [53], who do not find IOR in object-based coordinates, and other experiments have found IOR in location-based coordinates across object motion before [106, 107]. To facilitate the comparison between models with a space-based and with an object-based implementation, we define a mixed model *M.ll*, which is identical to *O.ll* (cf. Fig 2) but uses Eq (1) to calculate the inhibition $I$ instead of Eq (2).

The feature map $F$, the visual sensitivity map $S$, and the inhibition $I$ of targets (represented by objects or pixel locations) are multiplicatively combined at every time step to influence the decision of where to saccade next.

**Decision-making (IV).** We describe human scanpaths as a sequential process of saccadic decisions of either moving towards potential targets or staying at the currently foveated target [23, 108]. Such a perceptual decision-making process is typically described as a noisy accumulation of evidence from a given stimulus, which can be quantified using drift-diffusion models (DDM) [109–111]. The DDM is a well-established tool for explaining the cognitive and neural processes during decision-making [112–114]. It is based on the assumption that evidence for each choice is accumulated over time (drift), while each drift process is perturbed by random fluctuations (diffusion); a decision is reached as soon as sufficient evidence for one alternative is accumulated [110]. The DDM, by design, does not only model the perceptual choice but also the time until a decision is reached.

Since DDMs are typically applied to two-alternative forced choice tasks, we extend the evidence accumulation model to multiple options. Each potential saccade target is represented by a drift process with a time-dependent rate $\mu_i(t)$ and a noise strength $s$. As soon as one process, representing one potential target, reaches the decision threshold $\theta$, a saccade is executed towards this target.

The drift rate of a target $\mu_i$ is determined for each frame as the product of the scene features $F$, the visual sensitivity $S$, and the inhibition $I$ at the respective target based on the scanpath history. In the space-based models, we treat every pixel location $i \in [1, X \cdot Y]$ as a potential saccade target. Hence, the drift rates are computed as

$$\mu_i(t) = \mu(x, y, t) = F(x, y, t) \cdot S(x, y, t) \cdot (1 - I(x, y, t)). \tag{3}$$

In the object-based models, the objects in the scene define the potential saccade targets, as suggested, e.g., by Ref. [44]. Each object mask $O_i$ has the same dimension as a video frame, with entries within the object mask being 1 and outside the object being 0. To account for the different sizes of objects, we adapt the common practice to scale each object's perceptual size logarithmically (cf. [83]). The drift rate for each object $\mu_i$ is then given by

$$\mu_i(t) = \frac{\sum_{x,y} F(x, y, t) \cdot S(x, y, t) \cdot O_i(x, y, t) \; \cdot \; (1 - I_i(t))}{\sum_{x,y} O_i(x, y, t)} \cdot \log \sum_{x,y} O_i(x, y, t), \tag{4}$$

where the scalar $I_i(t)$ of a purely object-based model is replaced with the spatial map $I(x, y, t)$ in the case of the mixed model $M.ll$.

Each target in this framework accumulates evidence in favor of moving the gaze from the current position to the respective target. The evidence is based on the target's drift rate $\mu_i$ and random fluctuations $\epsilon \sim \mathcal{N}(0, 1)$ from the diffusion term scaled with the noise level $s$. The accumulated evidence of a target is given by its decision variable $V_i$, which is updated as

$$V_i(t + \Delta t) = V_i(t) + v \cdot (\mu_i(t)\Delta t + s\epsilon\sqrt{\Delta t}), \tag{5}$$

where $v$ is the fraction of time within $\Delta t$ spent on foveation and not on saccades. Scaling the update with $v$ instead of using a smaller $\Delta t$ is preferred since this makes it sufficient to update the decision-making process once per frame, i.e., set the time resolution for the update to $\Delta t = 1$ measured in frames. As soon as the decision variable of one target $V_i$ reaches the decision threshold $\theta$, a saccade is triggered towards this target. To ensure that foveation durations can be predicted with higher temporal resolution than $\Delta t$, we interpolate linearly the exact point in time the threshold is crossed within a time step. The foveation event stops at this interpolated time, the decision variables for all targets are reset to zero, and a saccade event begins. For the duration of the saccade, no evidence is accumulated, as described by Eq (5). We scale the saccade duration $\tau_s$ linearly with the saccade amplitude $a_s$,

$$\tau_s = 2.7 \; {}^{\text{ms}}/_{\text{dva}} \; \cdot a_s + 23 \; \text{ms}, \tag{6}$$

with values taken from Ref. [55].

The different number of potential saccade targets in a space-based compared to an object-based model (the number of objects is much smaller than the number of pixels for any given scene) can be accounted for by choosing the free parameters $\theta$ and $s$ appropriately.

**Gaze update (V).**   The gaze position in our modeling framework is updated at every time step (i.e., video frame). As long as all decision processes are below the DDM threshold, no decision is made and no saccade is triggered. In between saccades, the gaze position of humans also changes due to fixational eye movements and, since we consider dynamic scenes, smooth pursuit, which describes the eye movements when the gaze is fixated on a moving object. In

the object-based models, this is implemented by calculating the shift of the center of mass of the currently foveated object between two frames and updating the gaze position accordingly. Consequently, the gaze follows the currently foveated target object in a smooth pursuit motion or stays fixated if the object does not move. In the space-based models, we do not have object information available. To simulate realistic gaze behavior and produce comparable scanpaths, we achieve smooth pursuit by updating the gaze position according to the optical flow, calculated using PWC-net [60]. In addition, we model fixational eye movements like oculomotor drift or microsaccades [56–58] by updating the gaze position in every time step according to a 2D random walk with $\Delta x, \Delta y \in \mathcal{N}(0, \sigma_D)$. To match the fixational eye movements of human observers across frames to the frame rate of 30 fps in the VidCom dataset, we choose $\sigma_D = 0.125$ degrees visual angle (dva). This Brownian motion, in addition to making the generated data appear visually more like human eye tracking data, also contributes to the variability in the simulated scanpaths by shifting the center of the gaze-dependent visual sensitivity (module II).

If the decision threshold $\theta$ in the DDM is reached for a target, a saccade is triggered, and the gaze location is set to be on the corresponding saccade target at the time the saccade duration ends. In a space-based model, the new gaze position equals the pixel location of the target. The exact landing position of a saccade at time $t_0$ in an object-based model is sampled from all pixel locations within the object mask $O_i$ of the target object, where the probability $p_i(x, y)$ of each pixel to be the next gaze position is proportional to the scene features $F$ and the visual sensitivity $S$ (inhibition is uniform within each object), with

$$p_i(x, y) \sim O_i(x, y, t_0) \cdot F(x, y, t_0) \cdot S(x, y, t_0). \tag{7}$$

**Model parameters and fitting.** Both model families have five free parameters, which are listed together with the fitted parameter values in Table 1 for the space-based models and in Table 2 for the object-based models. The object-based model parameters are the range $\sigma_S$ of the visual sensitivity (cf. module II), the amount of inhibition $\xi$ of the currently foveated object, the slope $r$ of the decrease of inhibition of return over time (both (III), see Eq (2)), the DDM threshold $\theta$, and the noise level $s$ of the diffusion process (both (IV), see Eq (5)). In the space-based implementation, only one parameter in module (III) changes. Since the inhibition $I$ in Eq (1) is not bounded by the outlines of objects in the scene, we specify the range $\sigma_I$ of the Gaussian-shaped inhibition of return around the previous fixation locations. The mixed model $M.ll$, which is object-based except for the space-based IOR implementation in module (III), has therefore the same parameters as the space-based models and is listed in Table 1.

**Table 1. Free parameters of the space-based and mixed models and their values which were determined by evolutionary optimization.** Columns "Fit (last gen.)" show the mean and standard deviation across all 32 parameter sets from the last generation of the evolutionary optimization (for a plot of the full parameter space, see the S3–S7 Figs). Columns "Fit (best)" show the parameter values that lead to the best fitness among all individuals and which are the parameter values used for the model comparison shown in Fig 4. The parameter for the size of the oculomotor drift is set to $\sigma_D = 0.125$ dva for all models.

| Par. | Description | S.ll (low-level features) | | S.hl (high-level features) | | M.ll (mixed model) | |
|---|---|---|---|---|---|---|---|
| | | Fit (last gen.) | Fit (best) | Fit (last gen.) | Fit (best) | Fit (last gen.) | Fit (best) |
| $\sigma_s$ | Range of sensitivity [dva] | $12.61 \pm 0.34$ | 12.51 | $11.08 \pm 0.70$ | 10.71 | $9.21 \pm 0.32$ | 9.09 |
| $\theta$ | DDM decision threshold | $0.384 \pm 0.022$ | 0.384 | $1.72 \pm 0.026$ | 1.715 | $1.042 \pm 0.023$ | 1.068 |
| $s$ | DDM noise level | $0.011 \pm 0.001$ | 0.011 | $0.092 \pm 0.001$ | 0.092 | $0.241 \pm 0.004$ | 0.247 |
| $^1/_r$ | Slope of IOR decrease [$\Delta t$] | $257.4 \pm 18.7$ | 253.3 | $290.9 \pm 6.9$ | 296.0 | $166.1 \pm 21.5$ | 170.6 |
| $\sigma_I$ | Range of IOR [dva] | $7.37 \pm 0.35$ | 7.79 | $4.52 \pm 0.04$ | 4.51 | $6.49 \pm 0.58$ | 6.06 |

**Table 2. Parameters of the object-based models and their values which were determined by evolutionary optimization.** For details, see caption of Table 1.

| Parameter | Description | O.ll (low-level features) | | O.cb (center bias only) | |
|---|---|---|---|---|---|
| | | Fit (last gen.) | Fit (best) | Fit (last gen.) | Fit (best) |
| $\sigma_s$ | Range of sensitivity [dva] | 14.41 ± 0.71 | 13.75 | 5.81 ± 0.09 | 5.81 |
| $\theta$ | DDM decision threshold | 2.361 ± 0.175 | 2.128 | 2.922 ± 0.050 | 2.973 |
| $s$ | DDM noise level | 0.237 ± 0.008 | 0.230 | 0.228 ± 0.016 | 0.236 |
| $^1/_r$ | Slope of IOR decrease [$\Delta t$] | 232.3 ± 51.5 | 272.0 | 189.8 ± 50.4 | 159.4 |
| $\xi$ | Object-based inhibition | 0.66 ± 0.05 | 0.72 | 0.95 ± 0.01 | 0.95 |

The influence of model parameters and the predicted scanpaths were explored in extensive grid searches (see S1 and S2 Figs). The decision threshold $\theta$ of the DDM directly influences how long it takes to reach a decision and controls the mean duration of foveation events. With a low DDM noise level $s$, multiple simulations for the same scene lead to similar scanpaths, while a higher value of $s$ leads to a higher scanpath variability across different simulations. The Gaussian spread $\sigma_S$ of visual sensitivity determines the effective field of view around the current gaze position and strongly influences the average saccade amplitude. The slope $r$ of the decrease of inhibition over time determines the frequency of saccades that return to a previous target. The strength $\xi$ of the inhibition in the object-based models and the size $\sigma_I$ of the inhibition in the space-based models determine how often saccades are made within the same object as well as the frequency of saccades with short amplitudes.

Our framework was designed in a way that makes it easy to implement and prototype models (cf. Sec. *Framework architecture*) that reflect competing hypotheses. In this work, we compared five different models (*S.ll*, *S.hl*, *O.ll*, *O.cb*, *M.ll*), which have different assumptions about the role of objects and saliency information for attentional guidance. To evaluate these, we need an objective way of determining the optimal values of the free parameters for each model. Here we made use of the evolutionary optimization algorithm implemented in *neurolib* [115]. We used a Gaussian mutation algorithm with adaptive step size [116]. We optimized the similarity between simulated scanpaths and the human ground truth data with respect to two summary statistics: the distributions of saccade amplitudes and foveation durations. The corresponding fitness function $\mathcal{F}$ maximized by the evolutionary optimization is given by

$$\mathcal{F} = -\frac{d_{SA} + d_{FD}}{2}, \tag{8}$$

where the distance measures $d_{SA}$ for the saccade amplitudes and $d_{FD}$ for the foveation durations are given by the two-sample Kolmogorov-Smirnov (KS) statistic $D = \sup_x |F(x) - G(x)|$ of the respective distribution $F$ compared to the ground truth distribution $G$ from the human data. The parameters of each of the four models were optimized independently across 50 generations, using an initial population size of 64 and an ongoing population size of 32 individuals with the default settings of the evolutionary optimization (Gaussian mutation and rank selection with adaptive step size). The initial parameter ranges for each model are shown in S3–S7 Figs. We optimized the fitness function, Eq (8), simultaneously for all videos in the training set. We evaluated out-of-domain, i.e., on a different metricS than what we optimized the parameters for, how realistic the simulated scanpaths are for a given input video with a focus on analyzing which objects are selected for foveal processing and what function these foveations fulfill, as proposed in Ref. [117].

**Framework architecture.** We adopted the software structure (Fig 3a) of the *neurolib* framework [115]. Scanpath models in the *ScanDy* framework inherit from the `Model` base

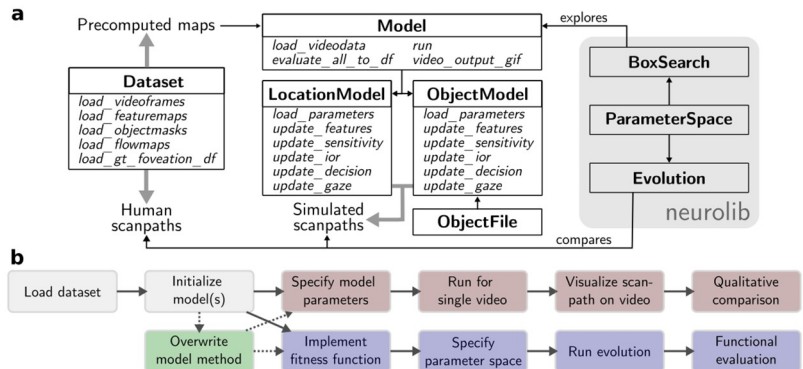

**Fig 3. Overview of the *ScanDy* architecture.** Software architecture of the *ScanDy* framework with example use cases. (a) Every box is a class with the name in bold and the most important methods in italic. The `Dataset` class provides the human scanpath data and makes the precomputed maps of the video data available for the models. At the core of the framework is the `Model` base class, from which all models inherit their functionality. The derived model classes have methods that correspond to the modules described in Figs 1 and 2. Object-based models might use the `ObjectFile` class to integrate object information in an efficient way. The classes on the gray background are native to neurolib and are used within *ScanDy* for parameter exploration and optimization. (b) Possible use cases of our framework include, in increasing complexity, generating scanpaths for a single video (red), comparing how well different models capture human gaze behavior (blue), and extending on existing models to test new hypotheses (green, common starting points in light gray). We provide Jupyter Notebooks of these and more examples on GitHub.

class. Its functionality includes initializing and running model simulations and the evaluation and visualization of the resulting scanpaths. Derived classes are the `ObjectModel` and `LocationModel`. Models that assume object-based selection use the `ObjectFile` class for each object within the scene. Analogous to the concept of object files in visual cognition [47], an instance of this class is used as a representation in which information about an object is linked and integrated. Independent from the specifics of the model, the simulated scanpaths have the same format as the eye tracking data from human participants. The human data is handled by the `Dataset` class, together with the corresponding video data, the precomputed object segmentation masks, the saliency maps, and the optical flow data for each scene. Which of these precomputed maps are used as inputs to the scanpath models depends on the specific hypotheses to be tested. The model parameters are handled by the `ParameterSpace` class of *neurolib*, which can be used for parameter explorations using the `BoxSearch` class or model optimizations using the `Evolution` class.

In the following, we briefly describe three potential use cases (Fig 3b), which are provided as examples with the code, detailed explanations and visualizations in Jupyter Notebooks on *ScanDy*'s GitHub page. To simulate a scanpath for a single video using a predefined model (Fig 3b, red), we first load the dataset that contains the precomputed saliency maps and object segmentations. Next, we either initialize an object- or space-based model and select the saliency map and reasonable values for the free model parameters (cf. Tables 1 and 2). With the initialized model, we only have to specify for which video(s) and how many scanpaths (different random seeds) should be generated in the run command. We can plot the resulting gaze positions on the video frame or analyze the scanpath and the output of the individual modules of the model using built-in functions.

In our second example (Fig 3b, blue), we want to compare the simulated scanpaths of different models quantitatively. For fair model comparison, we must ensure that better model performance is not just due to better model parametrization. Consequently, instead of evaluating a model for a single set of parameters, we define a fitness function, as in Eq (8), and specify a

range for each free parameter that should be explored during the optimization. During the evolutionary fitting procedure, the simulated scanpaths for different parameters are compared to the human data, finding parameter sets within the specified space that optimize our fitness function. The models with optimized parameters can then be compared in different ways to infer which underlying model assumptions best describe the attentional process in humans. For example, we can calculate the time spent in different foveation categories for a model (cf. Sec. *Functional scanpath evaluation* in *Results*).

In the third example (Fig 3b, green), we demonstrate the framework's modularity and the ease of modifying existing models to reflect new hypotheses. As shown with the mixed model (*M.ll*) investigated in this work, we can assume the visual sensitivity module (II), the target selection in the decision-making process (IV), and the gaze update (V) to be object-based and only the inhibition of return (III) mechanism to be space-based. This is achieved by initializing an object-based model and overwriting its `update_ior` method such that the locations of previous saccade targets are inhibited in allocentric coordinates. The modified model still has all functionalities and can be analyzed as one of the predefined models (cf. examples one and two).

## Dataset

We compared the simulated scanpaths with human eye tracking data on high-resolution videos of natural scenes from the VidCom dataset (link to the video and eye tracking data: http://ilab.usc.edu/vagba/dataset/VidCom/ [54] that were recorded under free-viewing conditions (i.e., no task instructions). We chose this dataset due to its high-quality human eye movement data (14 participants recorded with a 240 Hz infrared video-based eye tracker). The full dataset comprises 50 videos of everyday scenes, mainly showing individuals or groups of people, animals, or traffic scenes, each with 300 frames at 30 fps and without scene cuts. Participants were instructed to look at the center of the display at the beginning of each video. The scenes have high ecological validity, making them good candidates to explore and evaluate our modeling framework. About half of the videos in the VidCom dataset show a large number of objects with partially strong occlusion (e.g., people or cars behind fences or nets), which leads to failures of our object detection and tracking algorithms, and the videos had to be excluded. If more than 100 object IDs were assigned in a video, even after a first manual clean-up, the object masks were considered to be not useful and the video was excluded. The remaining 23 clips were randomly split into 10 videos for training, i.e. optimizing the free model parameters, and 13 videos to provide a sufficient statistic for testing how realistic the simulated scanpaths are. A list of the video names, the number of human participants for each video, and if it was used in the training or test set can be found in S1 Table.

We used the Deep EM classifier [118] for gaze event classification in human eye tracking data. When taking the average smooth pursuit duration (as classified using the algorithm by Ref. [118]) across observers for each video (all 10000 ms long), the smallest fraction of smooth pursuit we observe in a video is 543 ms (scene: *park01*). Across all videos, observers spend 2444 ms on average (median 2200 ms) doing smooth pursuit, compared to 5518 ms (median 5596 ms) of fixation. The classification between smooth pursuit and fixation events is noisy, however, even with state-of-the-art methods (with F1 $\approx$ 0.7 for smooth pursuit in [118]). Since both keep the current target foveated and are bounded by saccadic decisions, we combined smooth pursuit and fixations to foveation events. We evaluated for each point in time if the gaze position is on the background or within a 1 dva radius (to account for tracking inaccuracies or drift) of an object mask. To allow further analysis (see Sec. *Functional scanpath evaluation* in *Results*), we assigned every foveation event the ID of the object mask it spent the most

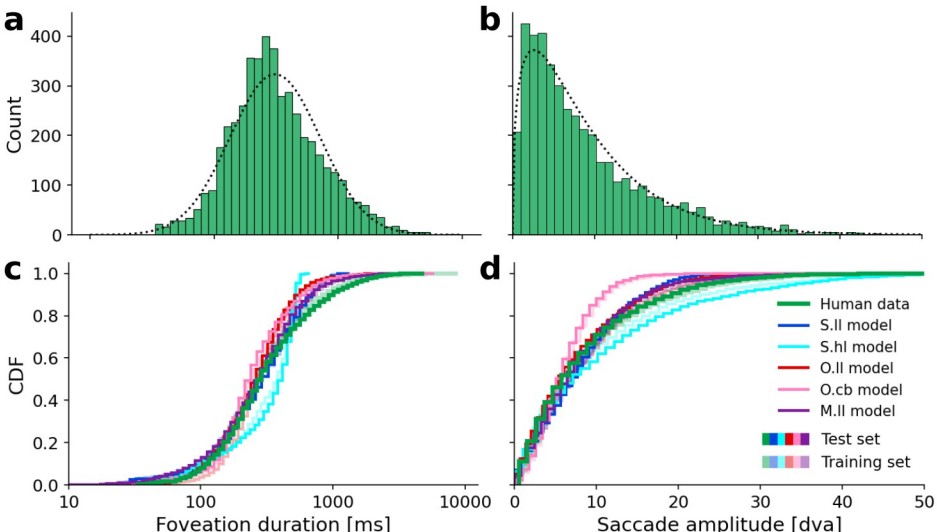

**Fig 4. Scanpath summary statistics of the human eye tracking data compared to the simulated model scanpaths.**
For each video in the dataset (10 in training, 13 in test data), we simulate 12 scanpaths (same parameters but with different random seeds) to roughly match the number of ground truth scanpaths. (a) Ground truth foveation duration distribution from all human participants across all videos. The dotted curve is a fitted log-normal distribution with $\mu = 5.735$ and $\sigma = 0.838$ (equiv. to an expected value of $e^{\mu+\sigma^2/2} = 439.6$ ms). (b) Ground truth saccade amplitude distribution. The dotted curve is a fitted Gamma distribution with shape $k = 1.43$ and scale $\theta = 6.50$ (equiv. to an expected value of $k\theta = 8.98$ dva). (c) Cumulative distribution functions (CDFs) of foveation durations are compared between human data (green) and the results of the space-based models with low-level features *S.ll* (blue), and high-level features *S.hl* (cyan), the object-based model with low-level features *O.ll* (red), and center bias only *O.cb* (pink), and the mixed model (object-based model with space-based IOR) with low-level features *M.ll*. The human data and modeling results from videos in the test set are represented by opaque lines, while transparent lines represent the corresponding training data and results. (d) Same as (c) but for saccade amplitudes. Model parameters were optimized using the evolutionary optimization algorithm. Parameters are listed in Table 1 for the space-based and mixed models and in Table 2 for the object-based models.

time on. We excluded foveation events shorter than 33.3 ms, corresponding to the displayed time of an individual video frame (30 fps). We further excluded all saccades with amplitudes smaller than 0.5 dva, which we consider as microsaccades and hence as a component of fixational eye movements [57]. The resulting summary statistics of foveation duration and saccade amplitude of the extracted eye movements are shown in Fig 4a and 4b.

## Results

*ScanDy* is designed to implement and test hypotheses about attention allocation and eye movement behavior when viewing dynamic real-world scenes. Depending on the hypothesis, competing model assumptions for individual mechanisms can be implemented within the framework. Here, we explore the hypothesis that objects play an essential role in attentional guidance, by systematically comparing predictions of models that incorporate different combinations of object and saliency information to the ground truth of human eye tracking data.

Our first model *S.ll* is space-based and uses low-level feature information [33] quantifying concepts from saliency and the feature integration theory [119] of visual attention (see Fig 1 with a, I, *left*). We primarily compare *S.ll* with the object-based model *O.ll*, which uses the same low-level saliency as scene features in module (I), but differs in all other modules (II-V) by the influence of objects on the attentional mechanisms (see Fig 2 with a, I, *left*). While the visual sensitivity (module II) is Gaussian in *S.ll*, it additionally spreads evenly across the

foveated object in *O.ll*, accounting for the spread of object-based attention [4, 6, 62]. The inhibition of previous saccade targets (module III) moves with the objects in *O.ll* [63, 105] compared to the Gaussian inhibition of previous target locations in allocentric coordinates in *S.ll*. The decision-making process (module IV) uses object masks as potential targets for saccadic selection [44, 45] and tracks the objects when updating the gaze position (module V) in *O.ll*, while *S.ll* selects individual pixel locations as saccade targets and updates the gaze with the optical flow to enable smooth pursuit behavior.

To account for the possibility that visual attention and target selection are, in fact, space-based but that the low-level features used in the first model do not allow for a sufficiently good saliency prediction, we introduce a second space-based model *S.hl* where the model architecture is identical to *S.ll* but that uses high-level saliency as predicted by TASED-Net [38] see Fig 1 with a, I, *right*). To better disentangle low-level saliency and object information, we add a fourth model *O.cb* that is object-based just like *O.ll* but does not use any scene features to compute the relevance of each object as a potential saccade target (see Fig 2 with a, I, *right*). It corresponds to the hypothesis that saliency is only relevant for eye movements through an object detection mechanism. With our fifth mixed model *M.ll*, we investigate the combination of object-based visual sensitivity, saccadic selection, and guidance, with a space-based IOR mechanism [53]. For this, we make use of the modular design of *ScanDy* by using the same model as *O.ll* but replacing module (III) with the space-based implementation (cf. Sec. *Framework architecture* in Methods). For a detailed description of the individual modules, see Sec. *ScanDy: A modular framework for scanpath simulation in dynamic real-world scenes* in *Methods*.

In the first part of this section, we show that all of these different models are capable of producing realistic scanpath statistics measured by the distributions of saccade amplitude and foveation duration. In the second part, we demonstrate that the object-based models, and in particular *O.ll*, lead to more realistic gaze behavior than the space-based models. This is assessed by analyzing how the simulated scanpaths move the gaze towards objects in the scene, based on functional foveation categories [117]. Complementary to the function of foveations —which are not sensitive to object identities—we also compared the absolute dwell times on individual objects of the simulated scanpaths and the human data. In the third part, we focused on the sequential aspects of the scanpaths by analyzing which objects are detected first and exploring the effect of the fixation history on upcoming saccades. By concentrating on phenomena related to IOR, we could investigate the individual contributions of attentional mechanisms. Our model comparison shows that typical IOR diagnostics cannot be explained through the implementation of the IOR mechanism (module III) alone but critically depend on the interaction between attentional mechanisms.

## Scanpath summary statistics

Eye movement events in human scanpaths on dynamic scenes have distinct statistics. The distribution of durations of all foveation events from all participants across all videos approximately follows a log-normal distribution (Fig 4a) with a median duration of 287.5 ms and, due to a few very long smooth pursuit events, a mean duration of 456.4 ms (expected value of the fitted distribution is 439.6 ms). The second important statistic of ground truth scanpaths is the distribution of saccade amplitudes (Fig 4b). The human data is well approximated with a Gamma distribution, which is consistent with the observed underrepresentation of short saccades and approaches an exponential distribution for large amplitudes. It replicates the mean amplitude of 8.98 dva.

Although we optimized the free model parameters to fit these two summary statistics (see Sec. *Model parameters and evaluation* in Methods), there was no guarantee that the models

with such a low number of parameters could actually reproduce these distributions. The log-normal shape of the foveation duration distribution is not explicitly implemented in the framework but had to result in the way how the evidence for moving the eyes is accumulated in the DDM (see Sec. *Decision-making (IV)* in *Methods: ScanDy*). Similarly, the Gamma distribution of saccade amplitudes can only be reproduced if the way of implementing the visual sensitivity *(II)* and the inhibition of return mechanism *(III)* allow for it.

The models with optimized parameters closely correspond to the distribution of the human foveation durations (Fig 4c). The cumulative distribution functions (CDFs) (plotted for the parameter configuration with the highest fitness for each model) provide an intuition on the distance between distributions as measured by the KS-statistic used in the fitness function Eq (8). (The KS-statistic between two distributions corresponds to the largest absolute difference between the two CDFs.) For both summary statistics, the CDFs of model results on the test data (opaque lines) closely correspond to the corresponding results on the training data (transparent lines). This shows that the basic scanpath characteristics of the models are, just as in the human data, robust to the exact scene content.

We observe that the distribution of human foveation durations is wider and has a longer tail than the model scanpaths. The *S.hl* model provides the worst fit and has a maximum foveation duration of 670 ms. In the human data, $\sim$ 19.33% of foveation events are longer than this, which is mainly due to long smooth pursuit events. The exponential distribution of the saccade amplitudes is well approximated by all models except for *O.cb* (Fig 4d). Since the center bias always attributes high relevance to the middle of the screen, irrespective of the scene content, this model is strongly discouraged from exploring parts of the scene with higher eccentricities, leading to shorter saccades.

The parameter space of the last generation of the optimization for each model (S3–S7 Figs) shows that most parameters converge to a common value with low variance. This indicates stability and a desired robustness of the model predictions to small changes in the parameters. Since our fitness function $\mathcal{F}$ in Eq (8) does not include information about the choices of saccade targets, how fast a target is returned to only has a subordinate effect on the fitness. The time course of IOR, parameterized by $r$ is, therefore, not well constrained by the optimization objective. The fitness function $\mathcal{F}$ allows for a trade-off between a good approximation of the foveation duration distribution (low $d_{FD}$) or saccade amplitude (low $d_{SA}$). Depending on the model, we observe that the evolutionary parameter optimization can achieve a better overall fitness $\mathcal{F}$ by converging to parameters that primarily improve the fit in $d_{FD}$ (e.g., *O.cb*, with mean and standard deviation across the last generation of $d_{FD} = 0.074 \pm 0.005$ and $d_{SA} = 0.190 \pm 0.004$) or in $d_{SA}$ (e.g., *S.hl*, with $d_{FD} = 0.184 \pm 0.003$ and $d_{SA} = 0.056 \pm 0.004$), or balance out both contributions (e.g., *S.ll*, with $d_{FD} = 0.079 \pm 0.006$ and $d_{SA} = 0.082 \pm 0.005$). Parameter fits from the *O.ll* ($d_{FD} = 0.102 \pm 0.005$ and $d_{SA} = 0.050 \pm 0.004$) and *M.ll* model ($d_{FD} = 0.039 \pm 0.003$ and $d_{SA} = 0.048 \pm 0.003$) achieve the best overall fitness.

We conclude that both the object- and space-based models are capable of reproducing basic scanpath statistics of foveation duration and saccade amplitude.

## Functional scanpath evaluation

Next, we assessed to which extent different model implementations can reflect human exploration behavior during scene viewing. This requires a scanpath metric that is applicable to dynamic scenes. It should consider the timing of eye movements and smooth pursuit events and be sensitive to the function that an eye movement may subserve in the context of the previous scanpath history (see Sec. *Evaluation of simulated scanpaths* in Discussion for a detailed comparison with other potential metrics).

Here we evaluate the scanpaths of the fitted models based on four different functional categories of foveation events [117]: An eye movement can either explore the background of a scene (*Background*), uncover a new object for the first time for foveal processing (*Detection*), further explore details of a previously detected object by making a within-object saccade (*Inspection*), or revisit a previously uncovered object (*Return*). Depending on the foveated object ID (cf. Sec. *Dataset* in Methods), we assigned each foveation event to one of these four categories. This allowed us to characterize human scanpaths by analyzing what percentage of foveation time in the ground truth scanpaths was spent on each category (Fig 5a, *Human data*). The first foveation predominantly explored the background since participants were instructed to look at the screen center at the beginning of each video, which corresponded to the scene background in most cases. Participants then quickly moved away from the background and started exploring the scene by detecting objects. The percentage of detection events decreases over time as more and more objects in the scene are detected, and already detected objects are more often inspected and returned to. With longer presentation times, the ratio of time spent exploring the background vs. inspecting and returning to objects is nearly constant. Both the test (opaque lines) and training data (transparent lines) show this general trend, while specific events in the videos cause fluctuations. This overall trend is preserved in the time course of the foveation category percentage for the models (Fig 5a), even though they show strong differences in viewing tendencies. We find that the *S.ll* has the highest proportion of background exploration of all models across the complete stimulus presentation time. The initial peak of detection events in the models with space-based IOR (*S.ll*, *S.hl*, and *M.ll*) is shorter than in the human data, and the increase in inspections, especially in the *S.hl* and *M.ll* models, is faster and more pronounced. The latter suggests that an object-based IOR mechanism is necessary to prevent the immediate inspection of otherwise not inhibited parts of a detected object. The overall time course of the object-based models *O.ll* and *O.cb* approximates the human data well.

For each foveation category, we also calculated the ratio of time spent across the whole video duration (Fig 5b). Model data shows a strong similarity in the balance of different foveation categories between the training and test data. Together with the consistent prediction of basic summary statistics (Fig 4c and 4d), this suggests that the models are able to effectively adapt to unseen videos despite the wide variation in scene content. We find that the *S.ll* model

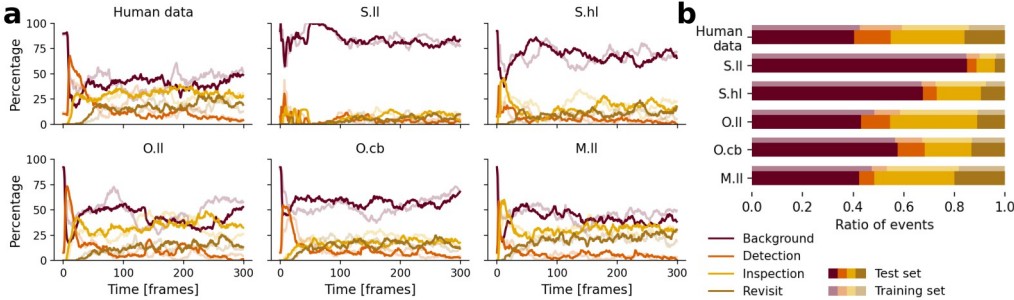

**Fig 5. Functional decomposition of scanpaths in foveation categories.** We distinguish the four functional categories "Background" (maroon), "Detection" (orange), "Inspection" (yellow), and "Returns" (khaki). Data plotted in opaque (transparent) colors represents the test (training) set. (a) Percentage of foveation events in each category across all human scanpaths and the model predictions as a function of time. Plotted is the proportion of scanpaths that are in each respective category. (Time spent during saccades or—in the case of the human data—periods in which the eye tracking data is corrupted by noise are not considered, and proportions are normalized to 100%.) (b) Proportion of time spent in each category averaged across all scanpaths from the human observers (data in panel (a) averaged over time) and for each of the five models averaged across all parameter configurations of the last generation of the optimization process.

spends more than 80% of the time exploring the background, which is about twice the time human participants spent on it. The models *O.ll*, *S.hl*, *M.ll*, and—surprisingly—also *O.cb*, match the time spent in each category more closely. We introduced a control condition where we manipulated the scenes such that the scanpaths no longer correlate with the object content. For this, we reversed the video frames in time and mirrored them on the x-axis and on the y-axis while keeping the scanpaths of the models and human observers unchanged. As expected, this manipulation leads to both the human and model scanpaths predominantly exploring the background and causes the previously described effects to disappear (we show the equivalent of Fig 5 for this control condition in S8 Fig).

To facilitate comparison, we expressed the time spent in each category for all five models relative to the human participants (Fig 6). It is apparent from the figure that all models show a significantly lower value for the time spent on detection events compared to the human data. This discrepancy is partially due to the fact that eye tracking data is corrupted by eye blinks or low data quality for more than 10% of the stimulus presentation time, while scanpaths simulated with the *ScanDy* framework do not assign any time to blinks or tracker noise, but only to foveation and saccade events. The resulting effectively longer foveation time does not contribute to more detection events in model scanpaths because each object in a scene can only be detected once. Again we do not find a qualitative difference in results on the training set (transparent) compared to the previously unseen videos of the test set (Fig 6).

The *S.ll* model (Fig 6a) spends most time exploring the background and only half the time exploring objects in the scene (see also Fig 5). In contrast to the *O.ll* model, the *S.ll* model cannot extract the parts of the scene that humans spend the most time looking at. Replacing the saliency map based on low-level features [33] with the TASED-Net [38] saliency prediction (i.e., model *S.hl*) leads to a significant improvement in the plausibility of the generated scanpaths (Fig 6b). This model spends more time exploring the objects, especially as inspections and Returns. The time foveating the background is reduced compared to the implementation with low-level features, though still $\sim 68\%$ higher than in the human data. This model spending less time foveating objects than the human scanpaths might be surprising since the high-level saliency model is highly sensitive to salient objects [38] and tends to only ascribe small

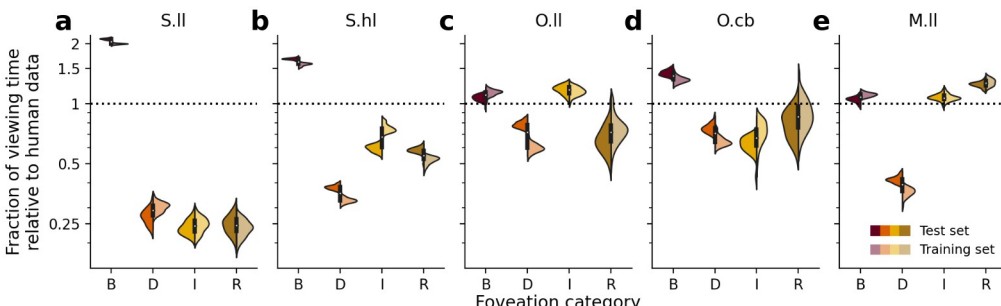

**Fig 6. Direct functional model comparison.** Violin plots of the ratio of the fraction of time spent in each foveation category predicted by the models and the fraction of time spent in each foveation category by the human observers. We show side-by-side the results for the videos of the test (left, opaque) and the training data (right, transparent) on a logarithmic scale. A ratio of one (dotted line) corresponds to a perfect prediction of how humans balance their exploration behavior. The four categories (**B**ackground, **D**etection, **I**nspection, **R**eturn) are shown on the x-axis. Individual data points for each model correspond to the 32 best parameter configurations (i.e., the last generation) of the evolutionary optimization and show the average value across twelve scanpaths per video in the respective dataset. Panels show the space-based models with low-level features *S.ll* (a), and with high-level features *S.hl* (b), the object-based models with low-level features *O.ll* (c), and with the center bias only *O.cb* (d), and the mixed model (object-based model with space-based IOR) with low-level features *M.ll* (e).

saliency values to the background. The high saliency values at locations that coincide with the object masks lead to a large number of short foveations within objects. The optimization objective, however, forces the model to also make long saccades away from these objects following the saccade amplitude distribution (Fig 4b), which—at least in part—leads to foveations on the background. If the gaze position moves far away from clusters of high saliency, the Gaussian decay of sensitivity means that it takes a long time until the decision threshold is reached again. Detailed inspection of the duration of foveation events in the background (median of 250 ms in the human data compared to more than 400 ms in *S.hl*) confirms that this model does typically overestimate the duration of background events. The detection events in the *S.hl* model are more strongly underrepresented compared to inspections and Returns, since the high-level saliency leads to a large number of very short foveations on objects and each object can only be detected once. In summary, this shows that although the high-level saliency map predicts well where people look on average, the space-based target selection leads to significantly different scanpaths in the *S.hl* model compared to the human ground truth.

In fact, saliency seems to play only a secondary role in the simulation of realistic scanpaths. Even without a saliency map, the object-based model *O.cb* produces a reasonable foveation behavior (Fig 6d). Unless objects are placed directly in the center, this model by construction assigns the majority of fixation time to the background of the scene, leading to an overrepresentation of $\sim$ 43% on average compared to the human data. The ratio of inspections is $\sim$ 37% lower compared to the ground truth, but the returns better match, on average, the human gaze behavior with $\sim$ 18% deviation. Detection events are reduced by about 25% of which about 10% can be explained through the larger effective foveation time compared to human data. The remaining discrepancy is due to shorter durations of the detection events, which could be adjusted with the object-based inhibition $\xi$ in Eq (2), but the parameters were not fitted on this metric. Although the "relevance" of objects in this model is only determined by their distance to the center of the display, it leads to a more balanced exploration behavior compared to both space-based models, *S.ll* and *S.hl*, which use state-of-the-art saliency models.

The average exploration of the background in the *O.ll* model is $\sim$ 7% higher than in the human data (Fig 6c). This slight overrepresentation is less pronounced compared to the other models and is likely due to situations where the distance between the gaze position and the closest foreground object exceeds the range of visual sensitivity, leading to slow evidence accumulation. The time spent on return events in both object-based models varies considerably between the different parameter fits (suggesting that the optimization objective in Eq (8) does not adequately constrain the inhibition of return parameters, which can be observed in the parameter space shown in S5 and S6 Figs) and is on average $\sim$ 32% lower compared to the human data.

The space-based IOR in the mixed *M.ll* model leads to an overrepresentation of return events of $\sim$ 25% compared to the human data. This occurs either due to only parts of the object being inhibited or through the object's motion, moving it out of the (in allocentric coordinates) inhibited region. The ratio of inspections is in good agreement with the human data, but due to an increased number of saccades within objects with short foveation durations, *M.ll* spends $\sim$ 58% less time on detection events than humans. Hence, the space-based IOR of *M.ll* does not lead to improvements over the *O.ll* model.

The functional scanpath analysis so far is agnostic to *which* objects in the scenes are detected, inspected, and returned to, such that—in principle—a model could dwell on very different objects than human observers but still perform well in this metric. To investigate this, we quantify the average time human observers foveate each object and compare it with the model prediction (see Fig 7). It is again important to note that we optimized the model

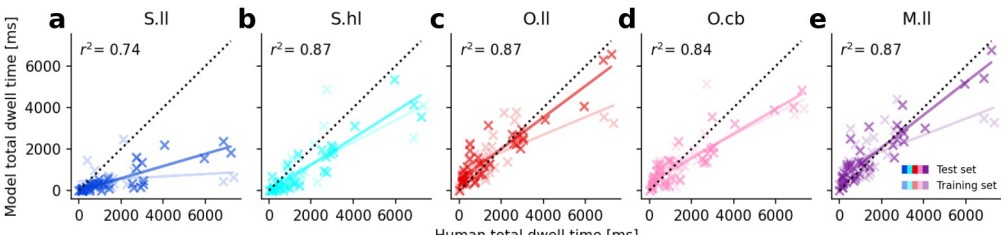

**Fig 7. Object-based model comparison.** Comparison of the average total dwell time of simulated model scanpaths compared to the average total dwell time across human observers for each individual object. We again average model predictions across parameter configurations of the last generation. (The results are qualitatively equivalent to averaging multiple model runs of the parameter configuration with the highest fitness.) A perfect prediction for how humans balance attention between objects would correspond to all data points lying on the dotted line with slope $m = 1$ and intercept $y_0 = 0$. We plot the objects of the test (training) set and the corresponding linear fits in opaque (transparent) colors for the *S.ll* model (a), *S.hl* model (b), *O.ll* model (c), *O.cb* model (d), and *M.ll* model (e).

parameters with a fitness function (cf. Eq (8)) agnostic to the object-based exploration behavior of humans.

The dwell times of the simulated scanpaths for individual objects as a function of the human data are for all models well described by a linear fit, with $r^2 \geq 0.74$ and $p \leq 10^{-12}$. Due to the overall excess exploration of the background in the *S.ll* model, the slope $m = 0.3$ of its regression line is below the desired value $m = 1$ (see Fig 7a). The same but less pronounced trend is also apparent in the *S.hl* model with $m = 0.64$ (see Fig 7b). The average dwell time across objects of the *O.ll* is similar to the human data. Its slope $m = 0.76$ is still below one due to an excess dwell time on objects which were detected by the model but not (or only for short times) detected by humans (reflected by the intercept $y_0 = 493$, see Fig 7c). Together with the overall underestimation of the time spent on detection events (see Fig 6), this shows that there are infrequent long detection events in the human data which are not captured by the models. The *O.cb* model prioritizes different objects only based on their size and proximity to the center but still has a better correlation with the human data than the *S.ll* model ($m = 0.59$, $y_0 = 358$, Fig 7d). The result of the *M.ll* model ($m = 0.79$, $y_0 = 486$, Fig 7e) is similar to the *O.ll* model.

We find that all models show a generally high correlation of object-specific dwell times with human observers. The models with object-based target selection (*O.ll*, *O.cb*, and *M.ll*) show an excess in the detection of objects (indicated by an intercept significantly different from zero), but *O.ll* and *M.ll* describe the absolute length of human dwell times on individual objects better than the spatial models (indicated by a slope closer to one).

In conclusion, the functional scanpath evaluation shows that we obtain the most realistic gaze behavior with the *O.ll* model, which is adding low-level saliency information to the object-based model and using it to prioritize between potential target objects.

## Sequential scanpath evaluation

In previous sections, we calculated measures across the entire scanpaths. Here we report complementary analyses, investigating the sequential aspects of the simulated and human gaze behavior. It is again important to note that we did not fit the model parameters to reproduce these metrics. Instead, we compared 12 runs of each model with the parameters leading to the highest fitness defined by Eq (8) (see Tables 1 and 2) with the human data (on average 12 observers per video).

A potentially interesting property of gaze behavior, which is not captured by our main analysis based on the functional evaluation, is the viewing order within a scanpath. Human observers and the models start their visual exploration in the center of the display (cf. Sec. *Dataset* in Methods). We then analyzed for each video which object the majority of observers detected first and compared it with the object that was most often detected first among the 12 runs of each model. We find that the *O.ll* and *M.ll* models, which only differ in their inhibition mechanism, both agree with the human data in 82.6% of the videos. In the remaining 17.4% of videos, there was at least one observer who also first detected the object preferred by these models. The *S.ll* model agrees with human data in only 69.6% of videos. This is less than the object-based model without saliency information, *O.cb*, with 73.9%. The *S.hl* model comes with 78.3% accuracy close to the *O.ll* and *M.ll* models but fails in some videos where the human data is unambiguous and all observers agree on the first object detection. Apart from the beginning of the videos, epochs of high inter-observer consistency also occur naturally as a consequence of the scene content. We show representative examples of such periods in S9 Fig, where we qualitatively compare the viewing behavior of the models with the human data. An additional qualitative comparison of full scanpaths simulated with the *S.ll* and *O.ll* models further demonstrates the more human-like target selection of the object-based model (simulated scanpaths as GIFs shown in S1–S4 Files).

We further analyzed the scanpath trajectories produced by different model implementations by comparing the relative angle between subsequent saccades in the simulated scanpaths with the human data (see Fig 8a). Consistent with previous findings in real-world scenes, we observe a prominent peak in saccades directing back to the last fixation location in the human scanpaths [27, 120]. This "facilitation of return" (FOR) effect [121] is not reproduced by the space-based models. Only 8.2% of saccades of the *S.ll* model have an absolute relative angle $|\varphi| > 135°$. The *O.ll* model, on the other hand, replicates this behavior despite an equally simple implementation of the scanpath history in its module (III) surprisingly well (39.8% with $|\varphi| > 135°$, compared to 36.9% in the human data). In contrast to their counterparts with low-level saliency, we do not find a particular turning angle preference in the *O.cb* and the *S.hl* model (27.8% and 27.2% with $|\varphi| > 135°$, respectively). The latter has a fitted spatial IOR extent that is significantly smaller compared to the *S.ll* model (see Table 1). Together with a more localized saliency map (cf. the example in Fig 1a), this makes high turning angles more likely. Interestingly, the model with object-based attention and target selection but space-based inhibition, *M.ll*, shows a high similarity with the *O.ll* model. Despite using the same scene features and having the identical implementation of module (III) as the *S.ll* model, the object-based selection of saccade targets leads to a realistic turning angle distribution (38.3% with $|\varphi| > 135°$).

The same viewing tendencies observed in the turning angle distributions are also reflected in the prevalence of return saccades, defined by $|\varphi| > 178°$ and an absolute difference in saccade amplitudes $|\Delta a_s| < 1.5$ dva (as in [120]). In the human data, 1.32% of all subsequent saccades fulfill this criterion, corresponding to an absolute number of 56 return saccades in all videos combined. Since there is only inhibition and no facilitation of return implemented in the framework, all models underestimate this value. Return saccades occur with 0.75% most often in the *O.ll* model, closely followed by *M.ll* with 0.73%. They occur less frequently for *O.cb* with 0.26% and almost never for the space-based models *S.hl* (0.13%) and *S.ll* (0.08%).

Next, we investigated the interdependence of the change in saccade direction and foveation duration. Forward saccades in the human data tend to be preceded by shorter foveations, while the preparation of saccades with larger turning angles is slower (see Fig 8b). This effect is strongly exaggerated in space-based models. It takes about twice as long for the *S.ll* model to accumulate evidence for a return compared to a forward saccade. This is a direct consequence of a broad and slowly decreasing inhibition suppressing the drift rate of the DDM for

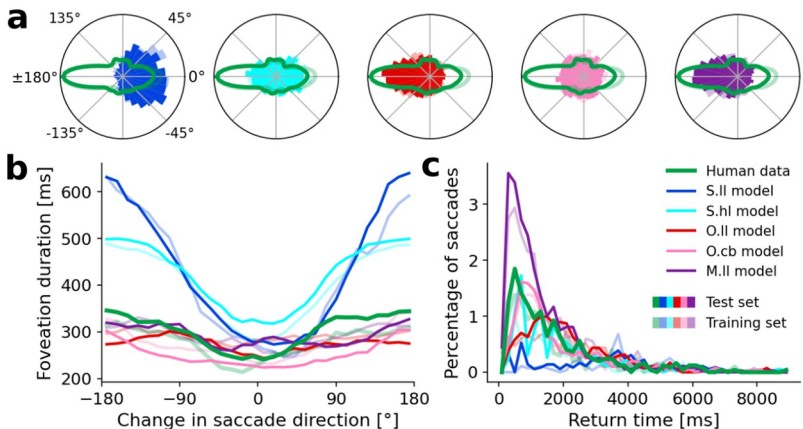

**Fig 8. Saccade statistics related to the scanpath history.** (a) Polar histogram of the relative angles observed between subsequent saccades. We show the binned distribution (12° bin size) of simulated saccades of each model in comparison to the kernel density estimate of the human data for the test (training) set in opaque (transparent). (b) Median duration of all preceding foveation durations of saccades for each bin (12° b.s.) for the human data and the simulated model scanpaths. We plot the median in such that a small number of long foveation events do not distort the statistics. To reduce fluctuations in the median, we apply a centered circular moving average across 5 bins. (c) Distribution of saccades that return to a previously uncovered object, depending on the time since the object has last been foveated. We normalized the distributions such that the y-axis shows the number of saccades returning to an object within each time bin (200 ms b.s.) divided by the total number of saccades.

previously explored parts of the scene (cf. Eq (3)). The same phenomenon occurs for *S.hl*, but not in the case of the *M.ll* model. Although the mixed model has a spatial inhibition map $I(x, y, t)$, it uses it to calculate object-based drift rates (cf. Eq (4)). This leads to foveation durations that—similar to the *O.ll* model—show no clear dependency on $\varphi$. The *O.cb* model, on the other hand, does show a similar effect as the human data with longer foveations before return saccades. The model comparisons for the number of return saccades, the relative saccade angle (Fig 8a), and its interdependence with foveation durations (Fig 8b) show that effects, which are typically ascribed to IOR, are actually primarily dependent on whether the selection mechanism for the decision-making process (module IV) is space-based or object-based. The actual implementation of the IOR mechanism (module III) and the salience (module I) also influence this behavior but only play a secondary role.

Finally, we also analyzed the time course of object-related return saccades (Fig 8c). In human scanpaths, 12.8% of all saccades return to a previously foveated object with a median return time of $\tau_r = 1200$ ms. This ratio is only 4.3% in the *S.ll* model ($\tau_r = 3482$ ms), while the more localized saliency map of the *S.hl* model leads to a ratio of saccades returning to objects of 9.3% ($\tau_r = 1556$ ms). Despite the similarities between the *O.ll* and *M.ll* models in direct return saccades, they largely differ in terms of object-centered return times. The object-based models approximate the percentage of saccades well (*O.ll*: 11.4%, *O.cb*: 12.4%) but show a delayed peak in return events (*O.ll*: $\tau_r = 1842$ ms, *O.cb*: $\tau_r = 1251$ ms). This delay is due to a shallow slope of IOR decrease (cf. Table 2), where a fitted $^1/_r = 272$ [$\Delta t$] for the *O.ll* model means that it takes almost the whole duration of a video (272 of 300 frames) for the inhibition of an object to go back to zero. A steeper slope $r$ would lead to faster return saccades without a large impact on the fitness $\mathcal{F}$ (as shown in the parameter exploration in S2 Fig and the large variance in $r$ in S5 Fig). The space-based IOR in the *M.ll* model tends to inhibit only parts of previously selected objects and does not track their movements, resulting in an overestimate of the prevalence of saccades returning to objects (22.5%) and shorter return times ($\tau_r = 967$ ms).

This is consistent with the overrepresentation of return events in Fig 6e. Hence, we find a significant effect of the IOR implementation in module (III) on the rate and the time course of object-related return saccades.

In summary, our analysis of sequential aspects shows that the *O.ll* and *M.ll* models best predict the prioritization between objects and exhibit similar behavior in terms of space-based IOR diagnostics but differ in their returns to objects. The model comparisons for the number of return saccades, the relative saccade angle, its interdependence with foveation durations, and the object-based return saccades suggest that phenomena typically associated with IOR actually arise from an interaction between multiple attentional mechanisms.

## Discussion

### Advantages and limitations of our mechanistic modeling framework for scanpaths in dynamic scenes

We present a general computational modeling framework for simulating human exploration behavior for ecologically valid scenes. By implementing psychophysically uncovered mechanisms of attentional and oculomotor control, *ScanDy* allows to generate sequences of eye movements for any visual scene. Recent years have shown a growing interest in the simulation of time-ordered fixation sequences for static scenes [14–17, 22, 23], as well as the frame-wise prediction of where humans tend to look on average when observing a dynamic scene [33, 37, 38, 40]. We are currently not aware of another computational model that is able to simulate time-resolved gaze positions for the full duration of dynamic scenes, analogous to human eye tracking data.

Scanpaths of multiple observers on dynamic scenes with gaze positions for each point in time are rich data, making their simulation a difficult task for any computational model. We achieve realistic gaze behavior in this work with mechanistic, well-interpretable models with only five free parameters. Achieving this simplicity and interpretability requires major simplifications and the here presented modeling framework does by far not contain all attentional mechanisms uncovered by psychophysical experiments.

One strongly simplified assumption we make in modeling the object-based visual sensitivity is that attention spreads instantaneously and uniformly across an object. Neurophysiological and behavioral studies have shown that attention spreads serially inside objects, which can be well described with a "growth-cone" model of perceptual grouping [122]. Especially since the currently established growth-cone model does not yet seem to generalize to real-world scenes [123], implementing and modifying it within the *ScanDy* framework seems to be a promising approach to investigate the object-based spread of attention.

Another simplifying assumption about visual sensitivity in the models presented here is that visual attention just spreads around the current gaze position. Experimental evidence, however, clearly shows that attentional resources are covertly allocated away from the current gaze position right before (e.g., [124]) and after (e.g., [125]) a saccade is executed. Extending the *SceneWalk* model [12] by such perisaccadic covert attention shifts improved the predictions of scanpaths, especially with respect to angular statistics [14]. The *ScanDy* framework can be extended analogously by including perisaccadic shifts in the sensitivity map $S$ (see Sec. *Visual sensitivity (II)* in *Methods: ScanDy*).

In this work, we assume a simple linear relationship between the amplitude and duration of saccades [55] and that saccades always precisely reach their target. For simplicity, we explicitly do not attempt to model the saccade programming and oculomotor control to execute this movement. There have been various models proposed in the literature for the velocity profile of a saccade based on its properties like amplitude and turning angle [126–130]. Including this

in our framework could be a valuable addition and might contribute to our understanding of perceptual continuity. We currently assume that objects can be tracked across time irrespective of the large and rapid shift of objects across the retina caused by saccades. A continuous and stable perception of the environment despite these shifts [131] could potentially be explained within our framework by modeling either attentional updating across saccades [124] or the incidental visual consequences of saccades [132], during which an object's retinal trace leads to motion streaks that can facilitate object correspondence across consecutive fixations [133].

So far, we only attempt to simulate scanpaths during free-viewing and hence do not include any top-down attentional control. In addition to bottom-up attentional guidance, there have been proposals for cognitive models of top-down visual guidance through task-driven attention [134, 135]. For future work, we expect that models within *ScanDy* could also be optimized to reproduce gaze behavior in visual search tasks by including guidance maps as feature map *F* (see Sec. *Scene features (I)* in *Methods: ScanDy*) based on the target's specific features or appearance [94, 136, 137].

Top-down attention in the form of expectations can also play an important role in guiding gaze during free viewing. It has, for example, been shown that already static cues implying motion robustly attract eye movements [138]. We currently have not incorporated any semantic representation of the scene into our framework. Our top-performing model (*O.ll*) relies solely on low-level features to prioritize between objects and to determine the specific location within that object to direct its gaze. Hence, the model does not capture aspects of human behavior guided by expectations or semantic preferences (e.g., towards faces as shown in S9 Fig). Such viewing biases could already be incorporated as additional feature maps into module (I) of the current framework, as described above. It may, however, be difficult to obtain such expectation or semantic meaning maps (cf. [139]) for dynamic scenes. Further utilizing objects as perceptual units, we plan on extending *ScanDy* to incorporate object-based semantic viewing preferences.

As a consequence of the small number of parameters, the model optimization within *ScanDy* is highly data efficient. This allows, in principle, for fitting individual differences by inferring model parameters for each human participant separately and systematically analyzing the variability in scanpaths [26]. The fixation tendencies between observers have been shown to vary strongly between individuals [140, 141]. We hope that with larger, high-quality eye tracking datasets of dynamic real-world scenes, *ScanDy* can become a useful tool to investigate individual differences in visual exploration behavior.

Our current model realizations within the *ScanDy* framework should be seen as easily modifiable baseline models. Depending on the specific interest in certain attentional mechanisms, models can be extended within the same framework. Analogous to our interest in the impact of object representations on attentional selection, we hope that the community can use *ScanDy* to implement different model variants based on competing assumptions and compare the resulting simulated scanpaths to evaluate hypotheses about gaze control and attentional selection.

## Evaluation of simulated scanpaths

The comparison of model scanpaths with human data is an essential way to validate our hypotheses and model assumptions. Scanpaths on static scenes $s_s$ usually consist of a short, ordered list of fixation positions $(x, y)$, with $s_s \in \mathbb{R}^{(2, N_F)}$, and $N_F$ being the number of fixations during the presentation time of the image. The saccade timing is typically not considered, since the scene content at each fixation does not change. This is, by definition, different in dynamic scenes. In videos, it is always necessary to specify when the gaze was at a certain

location since potential object and camera movement could influence what there is to see at this position. As a further complication, the gaze position can not only change due to saccades and (often ignored) fixational eye movements, as is the case for static scenes. Smooth pursuit plays an important role when observing dynamic scenes and in everyday life [142]. In this work, we combine fixation and smooth pursuit events, as both keep the current target foveated and are bounded by saccadic decisions. This means, however, that a foveation event in dynamic scenes does not have a single location, but that this location can (and often does) change over time. We, therefore, characterize a scanpath $s_d$ in dynamic scenes as a list of all gaze positions $(x, y, t)$ with a given time resolution, which means that $s_d \in \mathbb{R}^{(3,N_T)}$. Here we choose the number $N_T$, with $N_T \gg N_F$, of time steps equal to the number of frames in the video. The dimensionality of $s_d$, which is much higher compared to the dimensionality of $s_s$, gives rise to an immense variability in human scanpaths when observing dynamic scenes. Hence we do not expect the models to exactly reproduce these scanpaths, but to capture characteristic statistical features of it.

Thus far, scanpath simulation on videos is an under-researched topic, and this is reflected in the lack of scanpath evaluation methods that are applicable to dynamic scenes. To the best of our knowledge, there is no established metric to evaluate the realism of simulated scanpaths which captures both their spatial and temporal aspects and which does not assume static input scenes. In the following, we discuss metrics from video saliency prediction and from scanpath prediction on static scenes and elaborate on why they do not capture important aspects of the scanpath or are hardly applicable to dynamic scenes.

The overwhelming majority of work on modeling visual attention has treated attentional selection as a static problem, where gaze positions of multiple observers are described by a spatial distribution. There are multiple established methods to compare gaze data in the form of spatial distribution maps, as summarized by Ref. [143] for saliency predictions on images. These methods, however, typically ignore that attention is an inherently sequential process and do not consider any temporal aspects of the scanpath. There are extensions to also study spatiotemporal effects modeling fixation locations as arising from point processes [12, 144], e.g., by predicting the conditional probability of a saccade target given previous fixation locations. Calculating and evaluating the spatiotemporal likelihood function for all scanpaths and updating them in each frame would, in practice, already be computationally infeasible. But also conceptually, the duration of foveation events is typically not considered in these approaches (although an extension to labeled point processes by annotating points with their duration might be possible), and it is not clear how point processes might be generalized to include the change in gaze position during smooth pursuit events.

To evaluate video saliency models [35] or for quantifying the variability of human eye movements in dynamic scenes [26], frame-wise spatial distribution metrics have been used. The ground truth for these metrics for one video is computed by smoothing over all gaze positions from all human observers with a spatial or spatiotemporal Gaussian and creating a spatial ground truth distribution map for each frame. Due to averaging across different scanpaths, the resulting maps only provide information about where people are most likely to look, but not how individuals decide to move their eyes. The best-performing model under such a metric, like Normalized Scanpath Saliency (NSS), would consequently just stay foveated on the most salient part of the scene for the full duration of the video. Although humans might do that *on average*, in this work, we are interested in the way how humans actually move their eyes to explore the scene.

A well-established way to evaluate scanpaths on static scenes is using string-based metrics (e.g. [145]). The basic idea is to convert scanpaths to strings by defining a number of spatial

regions in the scene, for example, in the form of a checkerboard, which are represented by letters. Each fixation is then assigned the letter of the spatial region it lies in, such that the full scanpath can be represented by a string. Fixation durations can be taken into account by sampling the gaze position in fixed temporal bins. The comparison between two scanpaths can then be quantified using string edit metrics like the Levenshtein distance [146] or the Scan-Match method [147]. In dynamic real-world scenes, it is, however, not clear how the spatial regions should be defined. If the region boundaries are static, this means that the scene content within each region can change over time, making the encoding uninformative about what humans actually looked at. This problem can only be avoided if the region boundaries change over time to capture the movement in the scene, for example, by defining them based on objects. In this work, we chose a related approach to quantify how realistic the simulated scanpaths are.

To obtain realistic scanpaths, we prioritized reproducing the global statistics of human scanpaths in the model results. We showed that the simulated scanpaths capture the human foveation duration and saccade amplitude distributions. Such global statistics are, however, agnostic about where and when saccades are initiated *in relation to the scene content*. Since we used global scanpath statistics to fit the free model parameters, we expect all models to be able to reproduce the foveation duration and saccade amplitude distributions to a certain extent. A good free-viewing scanpath model should, when being tuned to reproduce these statistics, have implicit properties that lead to realistic exploration behavior given a visual scene.

In this contribution, we focused on investigating the role that objects play in attentional selection. It is, therefore, natural to use an evaluation metric sensitive to the extent and the particular way objects are visually explored. The approach for analyzing scanpaths proposed by Ref. [117] provides an intuitive way to quantify how and with what underlying function objects in the scene are selected. Although initially intended for static scenes, it naturally generalizes to dynamic scenes because it is based on objects. Importantly, we extended the three originally proposed categories (Detection, Inspection, and Return) by a fourth Background category. This addition effectively avoids a biased evaluation that would only consider foveation events on objects. This approach combines the benefits of string-based methods and dynamic region boundaries while additionally providing information about the extent to which objects are detected, inspected, or returned to. This evaluation is, however, not sensitive to the object identity, meaning a model could detect, inspect and return to objects ignored by human observers and still achieve perfect agreement. We controlled for this by directly comparing the simulated dwell time on each object with the human data. We complemented the functional evaluation, which is based on average viewing metrics, with sequential analyses that explicitly depend on the scanpath dynamics and previous foveation history. To incorporate the viewing order in our evaluation, we analyzed which objects are first detected when new scenes are explored. We focused this analysis on diagnostics of the IOR effect, including the relative angle of subsequent saccades, the interdependence of foveation duration and this angle, and the frequency and time course of object-based return saccades. This enabled us to investigate the individual contributions of object-based attention on different mechanisms. We additionally provide qualitative assessments in the form of an analysis of periods with high inter-observer consistency (S9 Fig) and animations of full scanpaths (S1–S4 Files).

## The role of objects in attentional selection

Our main interest was to use this computational framework to assess the importance of object-level attentional units for the guidance of human eye movements. Experimental research on object-based attention has repeatedly shown that objects play a central role in attentional

selection [6, 7] (cf. Sec. *Objects* in *Methods: ScanDy*). These insights did, apart from some notable exceptions (e.g., [49, 51, 52]), not yet translate into computational models, where the idea of spatial attention with saliency maps as the basis for attentional selection is still dominating. Our results demonstrate that models utilizing objects as the foundation for attentional mechanisms require only a small number of parameters to replicate human-like gaze behavior to a degree that cannot be achieved with space-based models of comparable complexity. The object-based model's ability to replicate a wide range of functional and sequential aspects of human scanpaths is especially surprising, considering that its parameters were fitted solely to match the general distributions of foveation durations and saccade amplitudes.

Specifically, we find that the *O.ll* model, which prioritizes objects based on low-level saliency information, performs best across metrics. This is consistent with results from scene viewing in static images, where Ref. [148] show that, when taking the preferred viewing location close to the center of the object into account, object outlines outperform models of early salience. Ref. [148] additionally use statistical models to estimate the impact of object properties on the fixation probability, finding that attention is primarily guided to objects and objects with higher low-level saliency are preferentially selected. Together with related work by Ref. [149], which also shows salience-based object prioritization, this emphasizes that previous work—while using complementary approaches—has come to the same conclusions as presented here.

The *O.ll* model, in particular, outperforms a space-based model with a high-level saliency map (*S.hl*), where object information may be contained in the scene features. The direct comparison between these models demonstrates that the improved performance of object-based models is not solely attributed to the *presence* of object information. Instead, it is the utilization of objects as perceptual units, enabling attentional mechanisms to act upon them, which leads to more human-like gaze behavior.

Our modular and mechanistic framework establishes a connection between the role of objects in gaze guidance and ongoing discussions about effect sizes in object-based attention [150] or the behavior of the IOR across eye and object movements [53]. We chose to concentrate on the latter aspect due to the suitability of dynamic scenes. For a mechanistic investigation, we compared purely space-based and purely object-based models with a mixed model *M. ll*, which only differs by its space-based inhibition in module (III) from the fully object-based *O.ll* model. In our functional scanpath analysis, we found more human-like exploration behavior from the *O.ll* model. In addition, we investigated sequential aspects of the simulated scanpaths, like the distribution of subsequent saccade angles and their interdependence with foveation duration, which are typically associated with IOR. We found that an object-based target selection in the decision-making process (module IV) actually has a stronger influence on IOR-related diagnostics than the implementation of the IOR mechanism in module (III). This result again points to the importance of object-based attentional guidance. Reaching a definitive conclusion on whether IOR is bound to objects or locations is not possible due to the limited number of return saccades in the here investigated dataset and the simplicity of our implementation. Evidence from static scenes suggests that instead of merely inhibiting previously fixated locations combination of saccadic momentum and facilitation of return (FOR) best describes human behavior [120]. A comparable approach has been implemented in the SceneWalk model [14, 15], which still uses IOR to prevent the model from only exploring the same high saliency regions, but combines it with FOR and perisaccadic attention (which can effectively act as saccadic momentum). We plan on extending the *ScanDy* framework in similar ways and applying it to a large video dataset with controlled camera- and object-motion in order to draw more definitive conclusions about the functional mechanisms behind phenomena related to IOR.

Our modeling results strongly suggest that objects are an important unit of perceptual selection and play a crucial role in guiding human gaze behavior. It is, however, not always clear what should be considered an object [64, 67]. We define humans, animals, and vehicles as objects, which are most relevant to the content in the here investigated scenes and can potentially show object motion. The remaining parts of the scene—including houses, furniture, trees, and other things—are assigned to a general background category. How exactly these object representations are formed in the brain is still an open question [151, 152]. We follow the argumentation of Ref. [7] and assume that object-based attention can act on fully evolved object representations already before they are selected as saccade targets. An interesting alternative to providing predefined object segmentation masks would be an implementation where object representations and attentional selection inform and influence each other. Optimizing object segmentation to provide realistic scanpaths would allow us to avoid defining what constitutes a visual object and could result in representations somewhere between locations and fully formed semantic objects.

## Conclusion

With *ScanDy*, we provide a modular and mechanistic framework for predicting scanpaths in dynamic real-world scenes. Despite its simplicity and the low number of parameters, our results show that models within this framework can reasonably reproduce spatial and temporal aspects of human scanpaths when watching natural videos. Comparing different models implementing competing model assumptions shows that the most human-like visual exploration behavior is obtained when object-based attention and object-based target selection are combined with saliency-based prioritization of saccade targets. *ScanDy* is an open-source project https://github.com/rederoth/ScanDy and due to its modular implementation, it can easily be extended and experimented with. We hope its availability will contribute to the field shifting further towards realistic environments and motivate more research on attention allocation in dynamic real-world scenes.

## Supporting information

**S1 Table. List of the 23 videos of the VidCom dataset used in this study.** The column "*Subjects*" shows the number of human observers, each contributing to ground truth scanpaths in the human data, and "*Split*" indicates if the video was used in the training or test set.
(PDF)

**S1 Fig. Influence of the free parameters in the location-based model with low-level features (*S.ll*) on scanpath summary statistics.** We use the mean parameter values from the last generation of the evolutionary algorithm, as reported in Table 1, as default parameters (indicated by the dashed line in the third and fourth rows). We then vary each parameter individually by multiplying with the factor 0.5, 0.75, 0.9, 1.1, 1.25, or 2. With all other parameters set to the default value, we simulate twelve scanpaths for each video in the VidCom dataset for each factor. From the resulting scanpaths, we plot the foveation duration (first row) and the saccade amplitude (second row) summary statistic as box plots, the resulting fitness measured by the respective KS-statistic (third row), and the fraction of stimulus time spent in each of the four foveation categories (fourth row).
(TIF)

**S2 Fig. Influence of the free parameters in the object-based model with low-level features (*O.ll*) on scanpath summary statistics.** We use the mean parameter values from the last generation of the evolutionary algorithm, as reported in Table 2, as default parameters (indicated

by the dashed line in the third and fourth rows). We then vary each parameter individually by multiplying with the factor 0.5, 0.75, 0.9, 1.1, 1.25, or 2 (the object-based inhibition parameter $\xi \in [0, 1]$ is set to $\xi = 1$ for factor 2). With all other parameters set to the default value, we simulate twelve scanpaths for each video in the VidCom dataset for each factor. From the resulting scanpaths, we plot the foveation duration (first row) and the saccade amplitude (second row) summary statistic as box plots, the resulting fitness measured by the respective KS-statistic (third row), and the fraction of stimulus time spent in each of the four foveation categories (fourth row).
(TIF)

**S3 Fig. Visualization of the parameter space of the space-based model with low-level features (*S.ll*).** The diagonal shows for each parameter the distribution across the last generation of the evolutionary optimization process. The last generation contains the 32 model parameter configurations with the highest fitness (as defined in Eq (8)). Other panels show how pairs of parameters relate to each other.
(TIF)

**S4 Fig. Visualization of the parameter space for the last generation of the space-based model with high-level features (*S.hl*), as in S3 Fig.**
(TIF)

**S5 Fig. Visualization of the parameter space for the last generation of the object-based model with low-level features (*O.ll*), as in S3 Fig.**
(TIF)

**S6 Fig. Visualization of the parameter space for the last generation of the object-based model with center bias only (*O.cb*), as in S3 Fig.**
(TIF)

**S7 Fig. Visualization of the parameter space for the last generation of the mixed model with low-level features (*M.ll*), as in S3 Fig.**
(TIF)

**S8 Fig. Functional decomposition of scanpaths in the foveation categories, equivalent to Fig 5, evaluated on altered scene content.** While the scanpaths from both the models and human observers remain unchanged, the corresponding video was reversed in time and mirrored on the x-axis and the y-axis. This manipulation significantly disrupts the correlation between scanpaths and objects in the scenes. Given that the majority of scene areas correspond to the general background category, we expected an extensive "exploration" of the background. (The relatively reduced background exploration time observed in human training data can be attributed to two videos featuring an object centrally positioned within the scene.).
(TIF)

**S9 Fig. Qualitative comparisons of viewing behavior of the models with the human scanpaths during periods of high coherence in the human data.** We identify prominent peaks in inter-observer consistency (measured by Normalized Scanpath Saliency (NSS) score as in [26]) and select three representative examples from the test set. We plot the current gaze position for the given frame for all observers and model runs and an arrow to the next saccade target. (Note that this might not reflect the actual trajectory due to smooth pursuit before the saccade.) The current gaze data of humans (in green) is plotted in all panels for comparison. (a) Video *park09*, frame 60. This example shows how object-based models tend to select the correct object but do not replicate the strong tendency of looking at faces observed in humans.

(b) Video *garden04*, frame 177. This is a failure case of the *O.ll* model, where not yet enough evidence is accumulated to select the object which newly entered the scene. (c) Video *walkway01*, frame 262.
(TIF)

**S1 File. Visualization of the *S.ll* modules for a single run.** Animation of the simulated gaze position (green cross) on top of visualizations of the different modules of the space-based model with low-level features (*S.ll*). The bottom left panel shows the original *field03* video. (I) Precomputed low-level saliency map with anisotropic center bias. Low values are shown in dark, high values in bright colors. (II) Gaze dependent Gaussian visual sensitivity map. Black means not sensitive (0), white means fully sensitive (1). (III) Inhibition of return map (value calculated for every pixel). White means no inhibition (0), black means fully inhibited (1). (IV) Visualization of the decision variable of each pixel-location. The saturation of a pixel represents the amount of accumulated evidence (white corresponds to 0, dark red to the decision threshold $\theta$). (V) The red circle indicates the next gaze position. The pixel values indicate the optical flow.
(GIF)

**S2 File. Multiple runs of the *S.ll* model.** Animation of twelve simulated scanpaths of the space-based model with low-level features (*S.ll*) on the *field03* video. Colors correspond to different random seeds when running the model with the parameter configuration with the highest fitness (see Table 1) and dotted lines indicate saccades.
(GIF)

**S3 File. Visualization of the *O.ll* modules for a single run.** Animation of the simulated gaze position (green cross) on top of visualizations of the different modules of the object-based model with low-level features (*O.ll*). The bottom left panel shows the original *field03* video. (I) Precomputed low-level saliency map with anisotropic center bias. Low values are shown in dark, high values in bright colors. (II) Gaze dependent visual sensitivity map, Gaussian with a uniform spread across currently foveated objects. Black means not sensitive (0), white means fully sensitive (1). (III) Visualization of the inhibition of return value of each object (attribute of the `ObjectFile` instance). White means no inhibition (0), black means fully inhibited (1). (IV) Visualization of the decision variable of each object (attribute of the `ObjectFile` instance). The saturation of the object mask represents the amount of accumulated evidence (white corresponds to 0, dark blue/red/green to the decision threshold $\theta$). (V) The red circle indicates the next gaze position. The pixel values indicate for each object how likely each position within each object is as a saccade target (calculated from the features (I) and sensitivity (II)).
(GIF)

**S4 File. Multiple runs of the *O.ll* model.** Animation of twelve simulated scanpaths of the object-based model with low-level features (*O.ll*) on the *field03* video. Colors correspond to different random seeds when running the model with the parameter configuration with the highest fitness (see Table 2) and dotted lines indicate saccades.
(GIF)

## Author Contributions

**Conceptualization:** Nicolas Roth, Martin Rolfs, Olaf Hellwich, Klaus Obermayer.

**Data curation:** Nicolas Roth.

**Formal analysis:** Nicolas Roth.

**Funding acquisition:** Martin Rolfs, Olaf Hellwich, Klaus Obermayer.

**Investigation:** Nicolas Roth.

**Methodology:** Nicolas Roth, Martin Rolfs, Klaus Obermayer.

**Project administration:** Martin Rolfs, Klaus Obermayer.

**Resources:** Klaus Obermayer.

**Software:** Nicolas Roth.

**Supervision:** Martin Rolfs, Klaus Obermayer.

**Validation:** Nicolas Roth.

**Visualization:** Nicolas Roth.

**Writing – original draft:** Nicolas Roth.

**Writing – review & editing:** Nicolas Roth, Martin Rolfs, Olaf Hellwich, Klaus Obermayer.

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
