## [Decision Letter · Decision Letter 0]

22 May 2023

Dear Roth,

Thank you very much for submitting your manuscript "Objects guide human gaze behavior in dynamic real-world scenes" for consideration at PLOS Computational Biology.

As with all papers reviewed by the journal, your manuscript was reviewed by members of the editorial board and by several independent reviewers. In light of the reviews (below this email), we would like to invite the resubmission of a significantly-revised version that takes into account the reviewers' comments. As can be seen from the reviewer's comments, they were on the fence as to whether this submission is suitable for PLOS CB. The major points raised seem addressable, however, with additional in-depth analyses and larger textual work. We would therefore like to invite a resubmission that addresses the points raised.

We cannot make any decision about publication until we have seen the revised manuscript and your response to the reviewers' comments. Your revised manuscript is also likely to be sent to reviewers for further evaluation.

Sincerely,

Tim Christian Kietzmann, Dr. rer. nat.

Academic Editor

PLOS Computational Biology

Marieke van Vugt

Section Editor

PLOS Computational Biology

Reviewer's Responses to Questions

**Comments to the Authors:**

Reviewer #1: This manuscript has many interesting bits and pieces but it does not have a coherent story.

Reading the abstract I stumbled over "Historically, eye movements were believed to be driven primarily by bottom-up saliency, but increasing evidence suggests that objects also play a significant role in guiding attention. " The view of an object is also bottom-up, stimulus driven. I view objects just as super features and not qualitatively different from other features. In the whole manuscript there is nothing about task / top-down guidance of eye movements (as is admitted later in the discussion). It is not just a isolated glitch, the topic reoccurs in lines 116 following and lines 174 following. This confuses with respect to the topic of the study. Precise phrasing throughout would be super helpful.

An important aspect of the study is the mechanistic model of guiding eye movements. That's interesting. It suffers somewhat from the suggested contrast of location based and object based models. As far as I understand it, it is all location based. Only two models get an extra object recognition neural network and segmentation algorithm. Well, a pure DNN can recognise objects as well and does a good job at predicting fixations on objects (e.g. DeepGaze 3). So, what exactly are the important differences between L.hl and O.hl? Is it that the network has to provide precise segmentation boundaries to guide fixations precisely on the objects, in contrast to the center of gravity of objects? Is O.hl just much better at detecting objects than L.hl? DeepGaze 3 does a pretty good job on static images. To my knowledge, it does not contain a specialised object model, it was just pre-trained on classification and then transfer trained. Wouldn't it work on videos as well? Then, you should obtain clear differences on the dynamic saliency maps. (btw. I'd like to see such data comparing models and humans, not just summary statistics that the model look at some object somewhere.) As this contrast relates to the objectives of the paper, it appears pretty important.

Then there is the inhibition of return (IOR) topic. I'm aware that IOR is a convenient mechanism that saliency map driven models do not always look at the same location. However, the early experimental evidence for IOR found a delay of saccades towards previously fixated locations, not a reduced probability to view previously fixated locations. Here, however, the term is used indiscriminately. Wilming et al., 2013, PLoS CB (sorry for citing my work) gives a reasonable account of this aspect. Given that it is a mechanistic model, here would be the chance for interesting investigations, e.g. does IOR in the correct sense of a delayed of revisits occur? What exactly is the distribution of saccadic vectors in subsequent saccades? How about the timing of fixation durations as a function of saliency of subsequent target? But such questions and timing is not addressed in the study.

Further, the study investigates eye movements on videos. This is interesting. However, fixations and pursuit movements are lumped together. There is no investigation on the influence of motion cues on the triggered eye movements. We do know that motion cues are effective even in static images (Acik et al, 2014, J Vis). How does the present model react to static cues implying motion? How important is the actual motion cues? Or an investigation whether IOR applies to moving objects or locations in allocentric coordinates. ... The study is performed using videos but falls short to deliver clear insights on this aspect.

The discussion contains a longish paragraph on the insufficiencies of measures comparing trajectories. But this is what the manuscript is about!? Why not use those measures we have? Not offering an alternative is not a solution.

The major analysis rests on the statistics of viewing objects/background, with the variation of detection, inspection, and revisits. Ok, interesting. But it leaves open whether the same objects are detected. The objects are viewed in identical sequence. In how far the results depend on the level of details of object labelling (e.g. forest or trees). Again, one model variant look at objects for a similar time, does this make it the better model.

Saccades modelled as instantaneous. Why? On static images I could understand but when you investigate videos it makes a difference whether a saccade last 40ms or 0ms. This simplification appears so unnecessary.

Looking back, there are many interesting bits and pieces but I fail to make up my mind what exactly I've learned. I'd recommend to reconsider what the main story of the study is and then go in depth for that story.

Signed, Peter König

Reviewer #2: Review for: Objects guide human gaze behavior in dynamic real-world scenes

Roth and colleagues present ScanDy, a novel computational model of human gaze behavior towards dynamic real-world scenes, validated against the VidCom dataset (Li et al., 2011). ScanDy predicts a series of foveation targets (fixations, pursuit) dynamically, i.e. in a time-resolved manner. This is a significant advantage over scanpath models for static scenes or frame-wise predictions of average fixation locations at a group level. It does so based on a modular combination of scene and observer features, allowing for flexible hypothesis testing.

The authors combine modules capturing salience, foveation and inhibition of return with a drift diffusion model of a decision process, leading to five free parameters. These are fitted to a human distribution of foveation durations and saccade amplitudes and then evaluated against human hold-out data (gaze towards a separate set of videos). Both, human behavior and model predictions are stable with respect to these basic parameters.

Importantly, model predictions are further compared to human behavior concerning the predicted proportion of background foveations versus three types of object foveations: Detection foveations (landing on a given object for the first time), Inspection foveations (saccades within a given object), and Returns (returns to a previously foveated object). This is done for four models: Two that base their predictions on locations (i.e. pixel-based foveation, IOR and decisions) and two that consider objects for foveation, IOR and the decision process. A model combining pixel-based low-level salience with object based foveation, IOR and decisions clearly outperforms the rest in recapitulating the proportions of Background / D/I/R foveations of human observers. The authors conclude that human gaze behavior towards dynamic scenes is (at least to a significant degree) guided by objects.

Strengths:

This paper is excellent. It is exceptionally well written, with a comprehensive literature review and thoughtful discussion. ScanDy is a substantial advance over previous models in the right direction (capturing natural gaze behavior) and promises to be a valuable tool for the community. Its modular architecture provides an avenue for testing numerous model features and hypotheses beyond the ones presented here. A particularly appealing feature is the accessibility of the code, including a Colab instantiation of an interactive notebook. Clearly, much effort and care went into this, and the result is impressive! The model also provides convincing evidence for the tested hypothesis that objects guide human free viewing to dynamic scenes beyond frame-wise observer averages.

Suggestions:

1) The authors evaluate model performance against the overall proportions of background / D/I/R foveations across all hold-out videos. The main message from the model comparisons seems to be that models lacking object-based information predict too few object foveations, providing convincing evidence that objects matter for foveations in dynamic scenes.

Nevertheless, it would be interesting to further challenge the model(s) by addressing the question of _which_ objects are being detected, inspected and returned to. The current evaluation scheme is agnostic towards this, so a model could (theoretically) achieve perfect agreement, even if there was zero overlap between the objects foveated by humans and the model. One way of going beyond this in the DIR framework would be to quantify DIR ratios (or absolute dwell times) separately for each object and average absolute deviations between predictions and observations across objects. This metric would complement the authors’ approach and provide an indication of the degree to which a model predicts _which_ objects attract D/I/R foveations.

2) The modular model architecture encourages readers to build upon it and explore their own ideas for extending or changing the model (some of which the authors foreshadow, such as perisaccadic shifts of attention). This is a fantastic feature and important to bear in mind regarding all questions about model specifications.

While hoping not to fall into the ‘why don’t you do it my way’ trap, I would like to point out that I find the choice of salience maps for the object and location-based models difficult to understand. Doesn’t the high-level salience map introduce object-based information (or a good proxy for that) to the location-based model? And why is the center-bias approach limited to the object-based models? Both these choices seem counterintuitive to me. If there are good reasons for them, it would be helpful to receive (further) explanation (apologies if I missed something obvious). If not, it may be more intuitive to leave out object-based information at this stage of the model altogether and keep the salience maps identical between the location- and object-based models (i.e. low-level and center bias maps for both, the location and object based models).

Minor:

1) The zero-lag of transitions between foveations causes excess total foveation times in the models. Is there a reason for not including a simple lag, scaling with amplitude to avoid this problem? The authors suggest more sophisticated extensions in this direction and I see how this would be more of a follow-up (see above). Still, I wonder whether a simpler solution wouldn’t go a long way to alleviate this problem and straightforward to implement in the existing model? Either way, this is just a suggestion and I would like to leave it to the authors whether to incorporate this or not.

2) The authors mention that objects in the context of their model are restricted to humans, animals and vehicles. It may make sense to highlight this more prominently and discuss the fact that all of these typically are moving objects. Cursory readers should be aware of the fact that ‘background’ in this context includes all sorts of inanimate objects.

3) I’m wondering: would it make any difference to drop the modelling of foveational eye movements?

4) Figure S7 (which I found informative) could be added as additional panels to Figure 5.

5) I’m not sure, I understand Figures S3-6. Does the diagonal show inter-individual parameter consistency? If so, could the lacking convergence for IOR decay point to genuine inter-observer variance?

6) I’m glad the authors found the D/I/R classification helpful. We have just published a preprint that introduces the approach and gives more detail than the cited abstract: https://psyarxiv.com/bqfdy/ Please note we now refer to ‘Returns’ instead of ‘Revisits’ to stay more consistent with the wider literature.

Ben de Haas

Reviewer #3: The question of eye guidance has been receiving a lot of attention in the modelling literature, probably starting from applying Koch & Ullman’s salience map to natural scenes in the late 1990s and not ending with current DNN-based models. Somewhat surprisingly, there have been largely distinct strands of modelling, such that it is a worthwhile endeavor to combine these into a single framework. Consequently, while none of the components of ScanDy may be groundbreaking in and by itself, the integration of many components certainly has a value for the field. This integration is performed by authors with the required in-depth knowledge of the field (and the various modelling strands). Hence, the paper in addition to an interesting model also provides a good overview on the state of the field, such that I would consider the manuscript potentially of interest also for the broad readership and therefore potentially suitable for PLoS CB.

I have one major concern, however:

(1) From the abstract I would have expected a direct comparison of scanpaths to human data (when reading it again, I understand that “including the proportion…” is meant as an exhaustive list). However, this happens only on the basis of summary statistics (those mentioned after “including…”). The authors put a lot of effort in the Discussion why they refrain from direct comparisons, but I don’t think these are fully convincing:

a. The authors could focus on periods/epochs of high inter-observer consistency with respect to the scanpath (e.g., nearly all observers looking at the same region at some point and then see how model and observers evolve from there).

b. They could apply their model to static scenes and compare the prediction to models tailored for this purpose with the metrics they discuss.

c. They could construct a control condition, where the distributions of figure 4 are still reproduced, but the scanpath statistics would differ (e.g., reversing or shuffling the video for computing the scanpath evaluation metrics)

d. They could include comparisons to static scanpath models (e.g., show the static model a frame from the video and apply the static prediction to the next 1s of the video).

There are probably more options, but prediction beyond approximately reproducing four coarse categories would strengthen the paper.

(2) One additional reason why I am worried about the scanpath evaluation is that to my understanding the assignment of Background, Detection, Inspection and Revisit heavily relies on the definition of an object. Could this bias in favor of object-based models.

Other comments (without particular order)

(3) In the context of extracting scene layout faster than the objects, the early work by Schyns and Oliva [e.g., 1] on hybrid scenes might be worth mentioning.

(4) Given that the authors find their O.ll model to be the best performing model and this is a central point of the present work, I would recommend to discuss the relation to the work by Stoll and colleagues [2]. These authors showed experimentally that a combination of hand-crafted object outlines and a central-bias like mechanism per object [3] outperforms other models, and – probably more relevant for the present discussion – also demonstrated that the selection among objects is then based on salience. While the approaches are very different to the present work, it seems that the conclusions regarding fixation probability in this work are similar to the conclusions on scan paths in the present work, such that these results nicely complement each other.

(5) Figure 4b: Even considering the “artifact” (l. 431), is it obvious that the reduced number of small saccades is all due to technical issues? Or might it be more appropriate to fit a Gamma-distribution, which for large amplitudes will approach the exponential?

(6) Given the relevance of smooth pursuit mentioned at several points: how abundant are smooth pursuit epochs in the dataset? Could the authors give an approximate fraction , like in XXX of the videos we observe smooth pursuit longer than – say – 300ms?

References

[1] Schyns, P. G., & Oliva, A. (1994). From blobs to boundary edges: Evidence for time-and spatial-scale-dependent scene recognition. Psychological Science, 5(4), 195-200.

[2] Stoll, J., Thrun, M., Nuthmann, A., & Einhäuser, W. (2015). Overt attention in natural scenes: Objects dominate features. Vision Research, 107, 36-48.

[3] Nuthmann, A., & Henderson, J. M. (2010). Object-based attentional selection in scene viewing. Journal of Vision, 10(8), 20.

**Have the authors made all data and (if applicable) computational code underlying the findings in their manuscript fully available?**

Reviewer #1: Yes

Reviewer #2: Yes

Reviewer #3: Yes

PLOS authors have the option to publish the peer review history of their article (what does this mean?). If published, this will include your full peer review and any attached files.

Reviewer #1: **Yes: **Peter König

Reviewer #2: No

Reviewer #3: No
---

## [Decision Letter · Decision Letter 1]

12 Sep 2023

Dear Roth,

We are pleased to inform you that your manuscript 'Objects guide human gaze behavior in dynamic real-world scenes' has been provisionally accepted for publication in PLOS Computational Biology.

Best regards,

Tim Christian Kietzmann, Dr. rer. nat.

Academic Editor

PLOS Computational Biology

Marieke van Vugt

Section Editor

PLOS Computational Biology

Reviewer's Responses to Questions

**Comments to the Authors:**

Reviewer #2: I thank the authors for incorporating my suggestions, especially the new figure 7 showing model fits for individual objects and for the updated explanations on model choices in the introduction. Overall: Congratulations on a great piece of work!

Reviewer #3: My concerns have been addressed sufficiently. I guess I still somewhat share the «bits and pieces» impression of reviewer #1, but to the credit of the authors they indeed have to cover a rather broad field, so even though I myself would have probably set a different emphasis here and there, it is understandable that they have to come up with some selection and the interested reader will have the option to directly use their model and data. So, I shall be happy with the present version.

**Have the authors made all data and (if applicable) computational code underlying the findings in their manuscript fully available?**

Reviewer #2: Yes

Reviewer #3: Yes

PLOS authors have the option to publish the peer review history of their article (what does this mean?). If published, this will include your full peer review and any attached files.

Reviewer #2: No

Reviewer #3: No

---

## [Editor Report · Acceptance letter]

5 Oct 2023

PCOMPBIOL-D-23-00435R1 

Objects guide human gaze behavior in dynamic real-world scenes

Dear Dr Roth,

I am pleased to inform you that your manuscript has been formally accepted for publication in PLOS Computational Biology. Your manuscript is now with our production department and you will be notified of the publication date in due course.

With kind regards,

Dorothy Lannert
